# Super-resolution proximity labeling with enhanced direct identification of biotinylation sites
Sanghee Shin[1,2,3,5,7], Song-Yi Lee[4,6,7], Myeong-Gyun Kang[4], Dong-Gi Jang[1,2], Jeesoo Kim[1,2], Hyun-Woo Rhee [2,4] ✉ & Jong-Seo Kim [1,2] ✉

Promiscuous labeling enzymes, such as APEX2 or TurboID, are commonly used in in situ biotinylation studies of subcellular proteomes or protein–protein interactions. Although the conventional approach of enriching biotinylated proteins is widely implemented, in-depth identification of specific biotinylation sites remains challenging, and current approaches are technically demanding with low yields. A novel method to systematically identify specific biotinylation sites for LC-MS analysis followed by proximity labeling showed excellent performance compared with that of related approaches in terms of identification depth with high enrichment power. The systematic identification of biotinylation sites enabled a simpler and more efficient experimental design to identify subcellular localized proteins within membranous organelles. Applying this method to the processing body (PB), a non-membranous organelle, successfully allowed unbiased identification of PB core proteins, including novel candidates. We anticipate that our newly developed method will replace the conventional method for identifying biotinylated proteins labeled by promiscuous labeling enzymes.

Proximity labeling is widely used to identify the spatial context of the proteome via genetically encoded enzymes using small molecules as tags to label the surrounding environment[1–4]. This technique is a promising alternative for spatial biologists to systematically map localized proteins in situ. APEX2[5,6], an engineered ascorbate peroxidase, and TurboID[1,7] (enhanced version of BioID), an engineered biotin ligase, use biotin-phenol (BP, biotin-tyramide) derivatives or biotin to tag surrounding proteins. Biotin-labeled proteins can be selectively isolated using streptavidin beads during liquid chromatography–mass spectrometry (LC–MS)-based characterization and identification[6,8]. Conventionally, biotin-labeled proteins are enriched at the protein level and eluted from the bead through an on-bead digest, in which eluted trypsinized peptides are mostly enriched, but unlabeled peptides correspond to biotinylated proteins[5,6,8] (Fig. 1a). As most eluted peptides from this conventional approach lack biotinylation site information, the enrichment results of the corresponding proteins are dependent on ratiometric approaches compared to their negative controls, which often provide false-positive (FP) results. Furthermore, the proximity labeling technique is well-known for its excellent efficiency in biotinylating and categorizing

proteins that are localized within membrane-enclosed regions, such as the mitochondrial matrix or lumen of the endoplasmic reticulum (ER)[5,7,9]. However, when the subcellular locals of interest for profiling localized proteins are relatively large or open spaces, such as the cytosolic face of the mitochondrial outer membrane or ER, conventional results frequently show insufficient portions of annotated true-positive (TP) proteins that are enriched compared to membrane-enclosed areas[10–12]. Several factors can contribute to this phenomenon, such as the labeling of non-specific proteins due to excessive enzyme activity or enrichment of secondary (or non-specific) proteins interacting with biotin-labeled proteins (Fig. 1a). Consequently, it is difficult to determine whether the unannotated enriched proteins are FP or TP using conventional approaches.

As separating unlabeled contaminants from the labeled protein sub-population during LC–MS analysis using the conventional approach is technically demanding, various techniques to identify biotinylation sites from biotin-tagged proteins have been developed that detect biotinylated peptides[13–16]. These techniques either focus on optimizing the protocol to

---

[1]Center for RNA Research, Institute of Basic Science, Seoul National University, Seoul 08826, Korea. [2]School of Biological Sciences, Seoul National University, Seoul 08826, Korea. [3]The Research Institute of Basic Science, Seoul National University, Seoul 08826, Korea. [4]Department of Chemistry, Seoul National University, Seoul 08826, Korea. [5]Present address: Department of Cancer Biology, Dana-Farber Cancer Institute, Boston, MA, USA. [6]Present address: Department of Genetics, Stanford University, Stanford, CA, USA. [7]These authors contributed equally: Sanghee Shin, Song-Yi Lee. ✉e-mail: rheehw@snu.ac.kr; jongseokim@snu.ac.kr

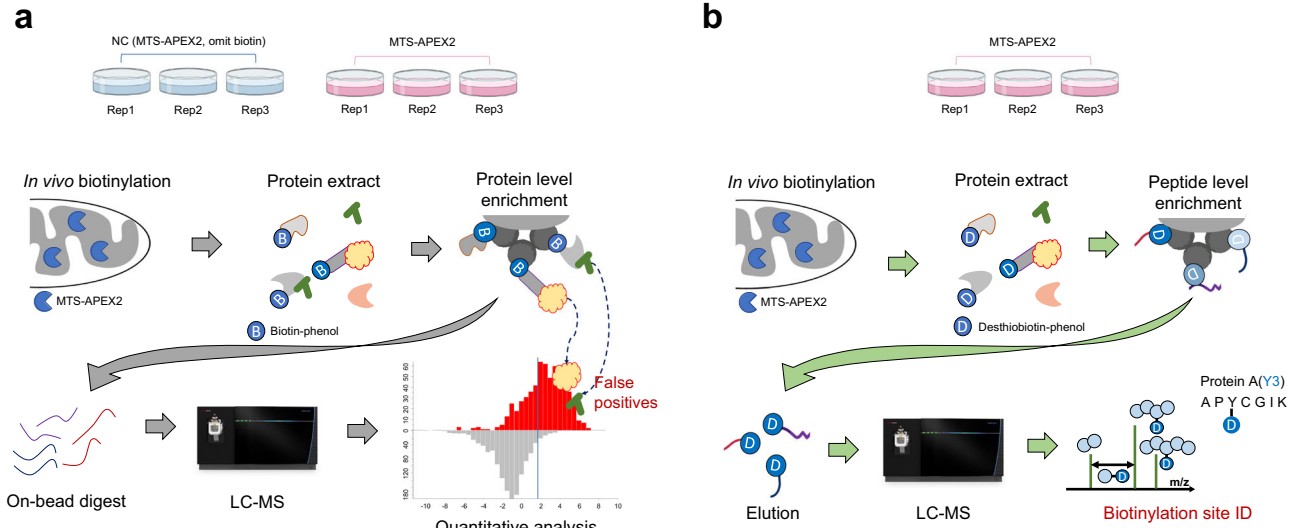

**Fig. 1 | Schematic workflow of conventional and enhanced direct identification of biotinylation sites. a** Schematics of the conventional workflow of biotinylated protein enrichment and identification through liquid chromatography–mass spectrometry. **b** Schematics of the biotinylated site identification method through enrichment of biotinylated peptides.

enhance the enrichment of biotinylated peptides or utilize an engineered avidin-like protein called Tamavidin 2-Rev[15].

Direct evidence of biotinylated sites can enable unbiased biotinylated protein identification, which can be labeled by engineered enzymes under physiological conditions. However, current approaches (e.g., Spot-ID[17], avidin-relative approaches[13,15], and anti-biotin antibody-based (Ab)[16,18] capture) are technically demanding and result in low yields. Therefore, we developed a newly optimized protocol that significantly improved the identification depth of biotinylated sites across the proteome and allowed for quantitative analysis under various conditions (Fig. 1b). Thus, our novel method for identifying biotinylation sites was directly compared with a conventional ratiometric approach as well as other related approaches to assess the practical advantages of using the novel method in biological applications. We demonstrated that in-depth identification of biotinylation sites significantly increases the accuracy and reliability of prospective protein identification based on proximity labeling in live cells and enables simpler experimental design. As demonstrated in our previous study, direct identification of biotinylation sites can provide information on membrane protein topologies where greater depth of identification can provide more useful data about determining the orientation of membrane proteins, including low-abundant membrane proteins. In this study, we applied the biotinylation site identification method for the first time to map a non-membranous organelle called the processing body (PB) with a suitable demonstration of an experimental design.

## Results
### Development of the super-resolution proximity labeling method and comparison with the conventional approach
The newly developed method was based on our previous protocol, Spot-ID[17], in which biotinylated proteins were captured using streptavidin beads and digested with trypsin. After removing the unlabeled peptides, biotinylated peptides were eluted from the beads using formamide, which was desalted prior to LC–MS analysis. However, this process was inefficient, as on-bead digestion often results in insufficient digestion and/or high background noise, including digested streptavidin, thereby diluting or masking signals of lower-abundance labeled peptides in the chromatograms. The additional desalting process reduced the yield of biotinylated peptides. As a result, the biotin-dependent enrichment method was optimized at the peptide level and eluted with an LC–MS-friendly buffer.

The protein extract from the lysate was precipitated to remove excess biotin and then resuspended for denaturation with 8 M urea buffer and then digested with trypsin. The digested sample was incubated with streptavidin beads to capture biotinylated peptides and then stringently washed. The enriched biotinylated peptides were eluted using the acidic organo-aqueous denaturation buffer[14] in which the acidic condition and the high percentage of organic solvent disrupt the integrity of the protein's structure[19]. The eluted sample was dried and directly used for LC–MS analysis, following resuspension in 25 mM ammonium bicarbonate buffer without an additional sample preparation process (Fig. 1b).

Direct identification of biotinylation sites within the peptide ensures that the corresponding proteins are labeled by engineered promiscuous labeling enzymes. Therefore, for applications in which the region of interest is a membrane-bound organelle, such as the mitochondrial matrix, we hypothesized that a negative control will not be needed to classify localized proteins that are biotinylated by proximity labeling techniques. We conducted a comparison experiment between the conventional ratiometric approach (Fig. 1a) and the newly developed biotinylation site identification method (Fig. 1b) based on the mitochondrial matrix proteome via APEX2 to analyze the efficacy of target protein identification and classification. In the conventional approach, mitochondrial matrix proteins of human embryonic kidney cells are labeled with APEX2 (mitochondrial targeting sequence with 24 amino acids, MTS-APEX2) using a negative control in which the biotinylation step is omitted. Mass spectrometric data analysis was performed using receiver operating characteristic-based analysis with a curated list[20] (MitoCarta 3.0) of TP and FP to generate an optimized cutoff threshold (Fig. 2a–c). Our new biotinylation site-mapping method used an MTS-APEX2-constructed cell line without any negative control. Mitochondrial matrix proteins were classified using biotinylation site identification, and TP proteins were curated with the MitoCarta database (Figs. 1b and 2a, d). All analyses were performed in triplicate. According to the results, 598 and 509 proteins were classified/identified from each conventional and biotinylation site-mapping approach, respectively (Supplementary Data 1, 2). Based on this result, TP proteins curated with the MitoCarta database showed 471 and 449 proteins, which corresponded to 78.8% and 88.2% of true positive rate (TPR) via conventional or biotinylation site-mapping approach, respectively (Fig. 2d–f). Between the two different approaches, curated TP proteins demonstrated good identification overlap (Fig. 2g). However, the overall quantitative composition of TP among all identified proteins within each dataset for the conventional approach was only 24% and 36% for the negative control and MTS-APEX2, respectively. Conversely, the newly developed biotin-site ID method showed 89% quantitative composition within each dataset corresponding to TP (Fig. 2h). Lastly, the remarkably low amount of streptavidin peptides from the biotin-

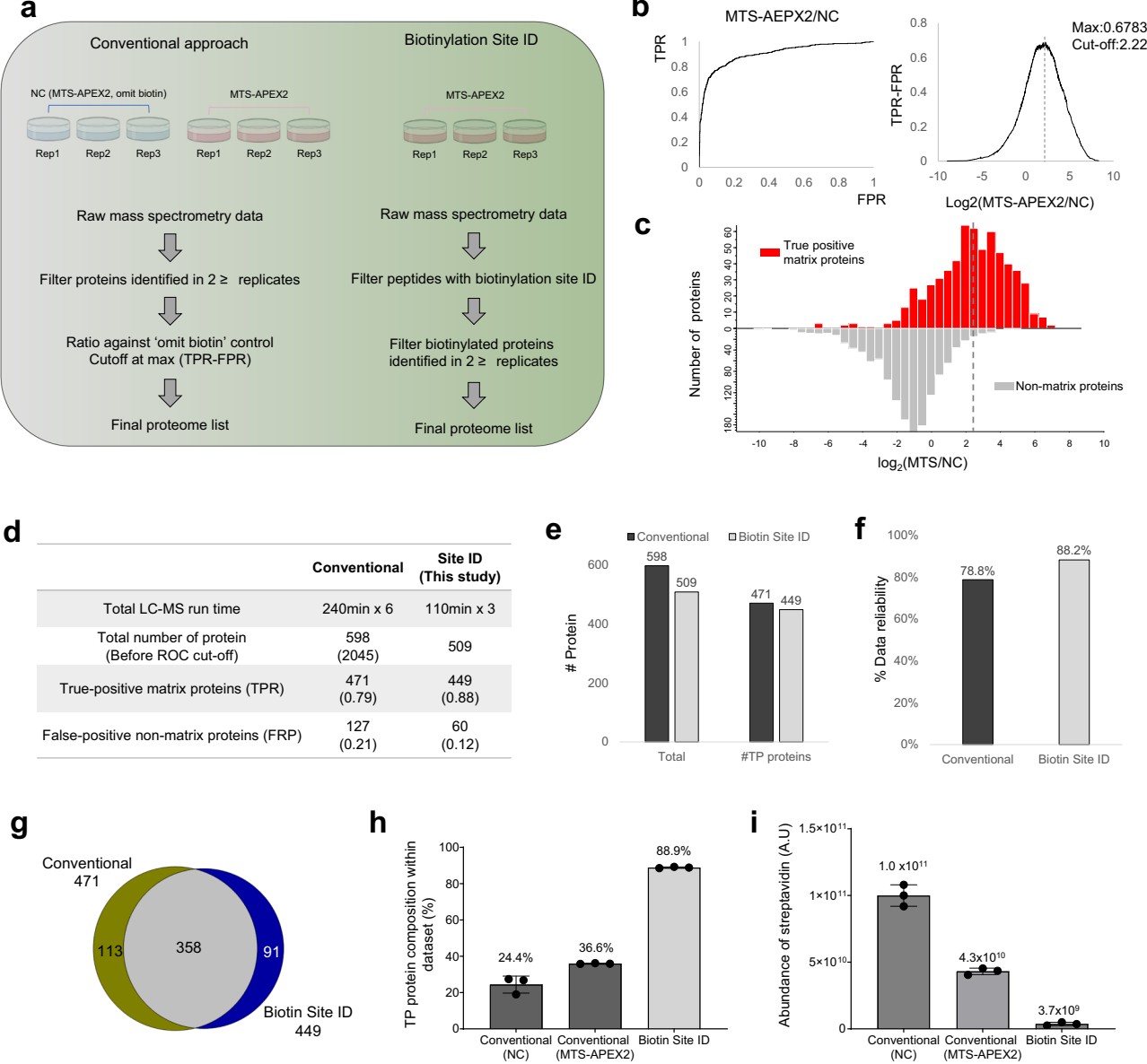

**Fig. 2 | Comparison of conventional workflow and super-resolution proximity labeling with enhanced direct identification of biotinylation sites. a** Filtering scheme for mass spectrometric data for conventional ratiometric approach (left) and biotinylation site identification method (right). **b** Establishing a cutoff threshold for conventional ratiometric approach. True positive (TP) rate (TPR) was plotted versus the false positive (FP) rate (FPR) in a receiver operating characteristic curve (left). TP proteins were curated using the MitoCarta 3.0 database. The optimal cutoff threshold was determined from the maximum value of TPR-FPR result (right). **c** Histograms showing the distribution of TP and FP proteins. **d** Detailed results of the conventional approach and newly developed biotin site identification method. **e** Total number of identified proteins from each method and a corresponding number of TP proteins. **f** Evaluation of data reliability. **g** Venn diagram of identified biotinylated proteins. **h** Composition of TP proteins among all identified proteins within a dataset of each method based on the quantified value (*n* = 3). **i** Comparison of streptavidin abundance measured from different approaches (*n* = 3).

site method in terms of peak area demonstrates that our biotin-site identification method can significantly reduce the contamination from background proteome (Fig. 2i). This demonstrates that the conventional approach can potentially increase data ambiguity by increasing FP when no database exists to curate TP for analysis.

The newly developed super-resolution proximity labeling method directly identified the biotinylation sites and outperformed the conventional ratiometric approach in terms of data reliability, simpler experimental design, LC–MS utility, and mostly unbiased identification of biotinylated proteins labeled with promiscuous labeling enzymes.

## Performance comparison among different biotinylation site-mapping approaches

The performance of the newly developed method for mapping biotinylation sites (Fig. 3a-3) was evaluated through a direct comparison between related approaches[16–18] [Spot-ID (Fig. 3a-1) and anti-biotin Ab (Fig. 3a-2)], which profiled the well-established context of the in situ-labeled mitochondrial matrix proteome (Fig. 3a, b). The developed method achieved sufficient results using only 25% of the initial sample amount required by Spot-ID and was comparable to the anti-biotin Ab approach. Moreover, the new method developed in this study achieved a 2-fold increase in biotinylated peptide spectrum matches per experiment compared to those with Spot-ID and a 1.6-fold increase with a 50% shorter gradient time for liquid chromatography with tandem mass spectrometry (LC–MS/MS) analysis than those with anti-

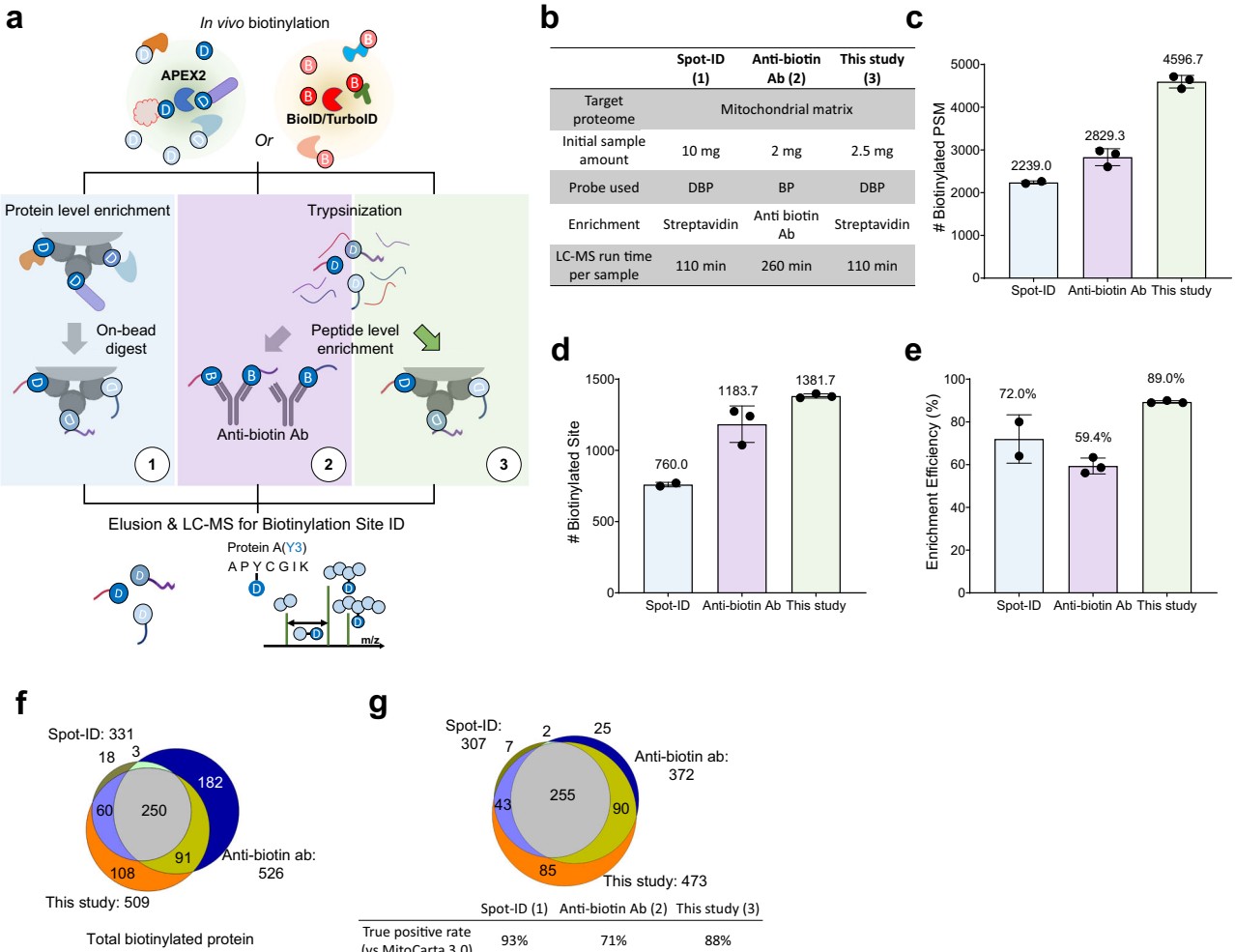

**Fig. 3 | Comparison of related approaches of biotinylation site identification.**
**a** Schematic illustrating the difference in related approaches of biotinylation site identification. **b** Summary table of related approaches for biotinylation site identification. Identification results of biotinylated peptide spectrum match (PSM) (**c**), site (**d**), and enrichment efficiency (**e**) of different approaches in biotinylation site identification methods. **f** Identification overlaps of total identified biotinylated proteins among different methods. **g** Identification overlaps of identified true positive proteins and their true positive rates among different methods. True positive proteins were curated using the MitoCarta 3.0 database.

biotin Ab (Fig. 3c). Thus, the results from this study yielded the highest number of biotinylation sites among related approaches for profiling the mitochondrial matrix proteome (Fig. 3d). Improvements toward obtaining higher efficiencies of biotinylated peptide enrichment yielded an enrichment efficiency of 89% with high inter-replicate reproducibility (Fig. 3e). This analysis demonstrated that our newly developed method had superb enrichment power for high-resolution site identification, resulting in the highest number of identified biotinylated peptides. Biotinylated proteins identified using different biotinylation site-mapping approaches had partially overlapping patterns (Fig. 3f). The filtered list of curated TP mitochondrial matrix proteins from each result demonstrated good identification overlap (Fig. 3g). This result suggests that the high number of proteins identified exclusively in the anti-biotin antibody (AbB) experiment may be owed to incorrect localization of the transfected APEX2 enzyme, which leads to a lower TP rate of identification. In addition, the Spot-ID approach and this study used desthiobiotin-phenol (DBP) as a probe that is more efficient than BP for LC–MS-friendly analysis of biotinylated peptides[17]. The AbB experiment used BP. Further experiments were carried out in-house to examine the practical performance of the AbB approach in comparison to avidin beads using a stable cell line expressing a mitochondrial matrix-targeted APEX2 construct with a BP probe. Mitochondrial matrix proteins were labeled with APEX2 using BP, and the BP-labeled samples were enriched using either streptavidin beads (SAB) or AbB. These results were compared with those of

our standard workflow, which was labeled using DBP and enriched using streptavidin beads (SAD) (Fig. 4a). Triplicate experiments revealed that SAB had ~50% more identified biotinylated peptides and 30% more identified biotinylation sites than AbB (Fig. 4b, c). When compared to the avidin bead-based enrichment approach, Ab enrichment demonstrated significantly lower enrichment efficiency of biotinylated peptides in tryptic digests of whole protein extract (Fig. 4d). These results reflect the quantitative reproducibility of the identified biotinylated peptides among replicates. The average reproducibility of AbB was the lowest for enriching biotinylated peptides, while that of SAD was the highest (Fig. 4e). Overall, streptavidin coupled with the DBP probe showed the best performance in terms of identification, enrichment efficiency, and reproducibility (Fig. S2a–f; Supplementary Data 3) of biotinylated sites on in situ biotin-labeled proteins, confirming its suitability for biological experiments.

We further crosschecked the elution efficiency of the acidic organo-aqueous buffer against that of the popular Laemmli buffer (2% SDS) based on the dot blot analysis with streptavidin-HRP due to the incompatibility of the Laemmli buffer for direct LC–MS analysis. The DBP-labeled peptide samples (by MTS-APEX2) enriched on streptavidin beads were used for the elution comparison. The results clearly showed that the acidic organo-aqueous buffer used in this study was significantly more efficient at elution than the Laemmli buffer (Figs. 4f, g, and S2g), in addition to being LC–MS compatibility.

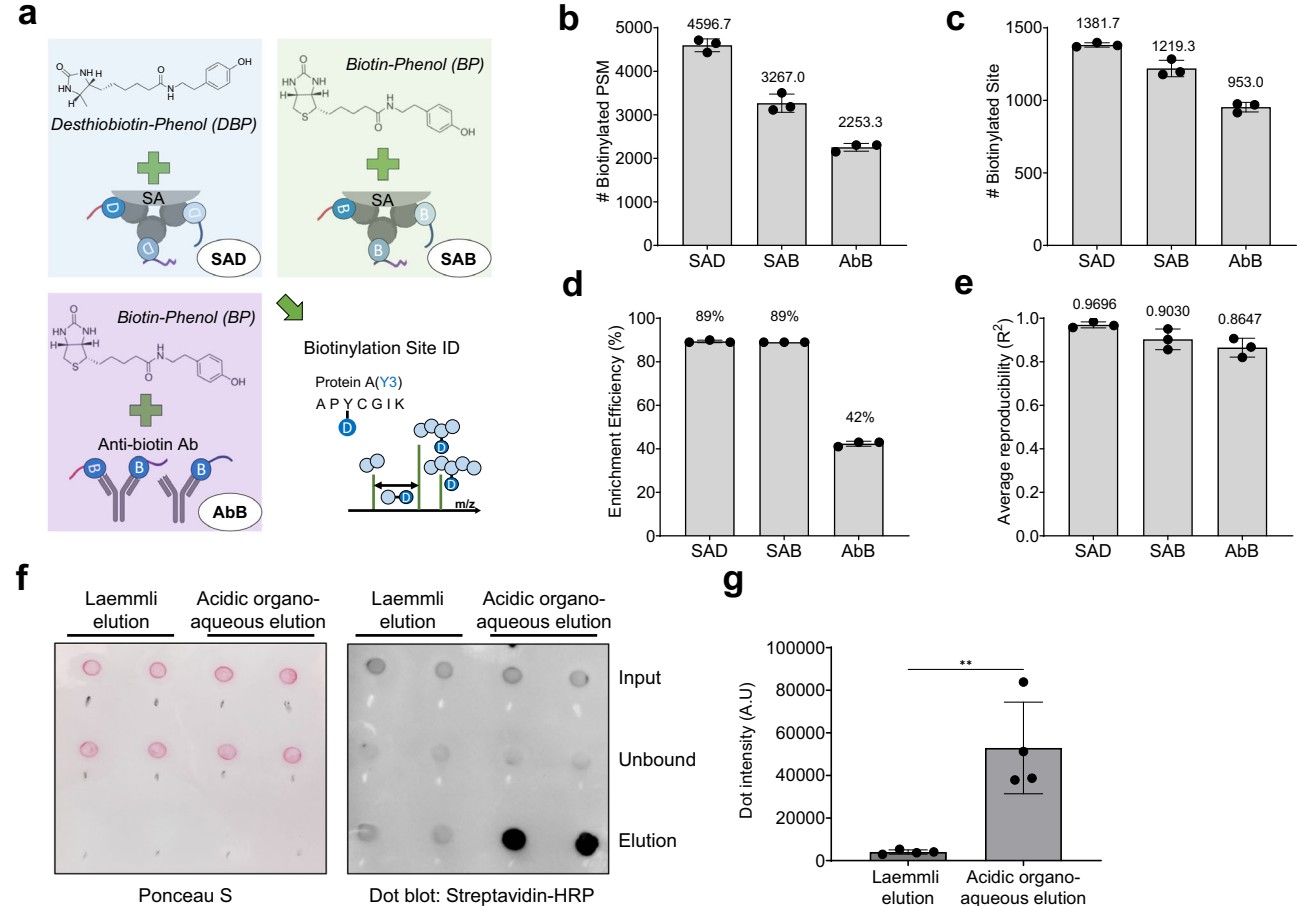

**Fig. 4 | Comparison of different probe-capture combinations for identifying biotinylated sites. a** Schematic illustrating the differences in probe-capture combinations for the biotinylated peptide enrichment strategy. Identification results of biotinylated PSM (**b**), site (**c**), enrichment efficiency (**d**), and average reproducibility among replicates (**e**) of different probe-capture combinations of biotinylated peptide enrichment approaches. **f** dot blot result by streptavidin-HRP for comparing different elution buffers. **g** quantitative comparison of dot blot result of eluted samples from Figs. 4f and S2g, $n = 4$, $p$-value < 0.004.

## Enhanced mapping of the mitochondrial matrix and inner mitochondrial membrane (IMM)/inner mitochondrial space (IMS)

To evaluate the improvements in the newly developed method and its applicability in biological contexts, we adopted the same context as in our previous study. The mitochondrial matrix and IMS were targeted using the MTS-APEX2 and SCO1-APEX2 constructs, respectively (Fig. 5a). In brief, 2624 biotinylation sites were successfully identified, representing a total of 830 proteins with 1901 biotinylation sites, representing 752 proteins reproducibly identified in at least two replicates (Supplementary Data 4). Although many of the biotinylated sites were included in previous results, a greater number were identified for the first time (Figs. 5b and S3a). Furthermore, we obtained quantitative information on all identified biotinylation sites using label-free quantification (LFQ), embedded in MaxQuant software[21,22], with high reproducibility (Fig. S3b, c). The biotinylation sites were divided into two distinct clustering groups based on these data, with groups 1 and 2 corresponding to biotinylation by SCO1-APEX2 and MTS-APEX2, respectively (Fig. 5c). Gene Ontology (GO) enrichment analysis of the cellular components of these groups showed good agreement with the target subcellular location of each bait (Fig. S3d), demonstrating the superior identification of biotinylated sites over previous studies (Fig. S3e).

Additionally, the inter-cluster overlap of the corresponding proteins was evaluated using clustering analysis. This revealed two exclusive groups that represented Matrix or IMS-resident proteomes and a small number of overlapping proteins biotinylated by MTS-APEX2 and SCO1-APEX2 (Fig. 5d). MTS-APEX2 targets the mitochondrial matrix, and SCO1-APEX2 targets the IMS, which is physically separated by the IMM. The proteins in the exclusive groups agreed well with the experimental context, with

overlapping proteins corresponding to IMM proteins, indicating that we had successfully defined the subcellular localization of mitochondrial proteins.

As shown in our previous Spot-ID study, biotinylation sites can indicate the topology of identified transmembrane proteins by matching the sites with corresponding APEX2-conjugated bait constructs that target suborganellar locals of interest. Using a new super-resolution proximity labeling method, we could identify more biotinylated sites than before, thus achieving greater benefits toward confirming membrane protein topology, including the identification of biotinylated sites in newly defined proteins. The increased number of identified biotinylated sites led to the discovery of more distinct sites in previously identified proteins, providing additional confidence in proposing a new topology for transmembrane proteins. For example, the proposed topology presented by Spot-ID (Fig. 5e) relied on biotinylation sites identified on only one side of the membrane, whereas our novel method simultaneously identified biotinylation sites on both transmembrane faces. Moreover, based on the results of our reproducibly identified biotinylation sites accurately reflecting the determined topology of 28 transmembrane proteins at the IMM, we inferred the topological orientation of 42 IMM transmembrane proteins that were not precisely defined (Fig. S3f–i).

## Enhanced mapping of PB proteome via proximity labeling

Translationally repressed mRNAs are stored in the cytoplasmic ribonucleoprotein (RNP) granule known as PB and associated proteins responsible for mRNA decay and storage[23–28]. It is challenging to isolate PB using the traditional fractionation method as it is a cytosolic non-membranous

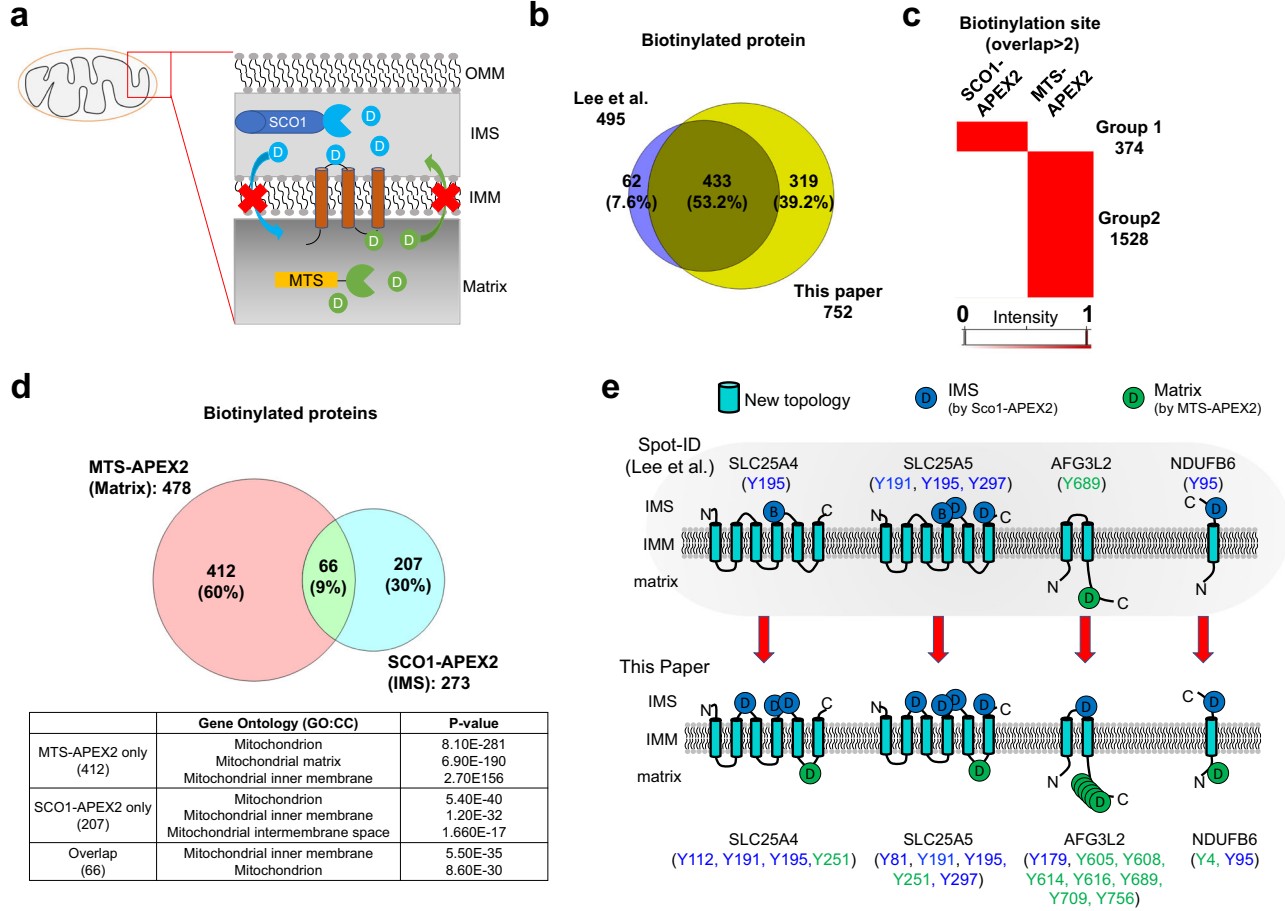

**Fig. 5 | Analysis of the mitochondrial matrix and intermembrane space proteome via super-resolution proximity labeling with enhanced direct identification of biotinylation sites. a** Schematic figure of biotin-labeling mitochondrial matrix, inner membrane, and intermembrane space proteome via proximity labeling using APEX2 construct. **b** Identification comparison is done between Lee et al.'s[13] and this study. **c** Heatmap of clustered identified biotinylation sites. **d** Overlapping and corresponding Gene Ontology terms of biotinylated proteins identified using MTS-APEX2 and SCO1-APEX2. **e** previously mapped topology of membrane proteins by Lee et al. (upper panel). Advanced identification of biotinylation sites applied on topology mapping by this paper (lower panel).

organelle that exhibits unique physiological compartmentalization from the cytosol, known as liquid–liquid phase separation[24,28–30]. Thus, we used APEX2 and TurboID to profile the PB proteome using our newly developed biotinylation site-mapping method for unbiased biotinylation protein-mapping.

DCP1a, LSM14a, and RCK (DDX6) proteins, which are known core PB components, were used as bait and conjugated to either APEX2 or TurboID (Fig. 6a). HEK293T cells stably expressing APEX2 and TurboID conjugates were evaluated for PB-targeting and biotin-labeling specificity using confocal fluorescence imaging, with the results showing good agreement (Fig. 6b). For enhanced mapping of biotinylated PB proteins, cytosolic APEX2-NES or TurboID-NES was used as a control to distinguish core PB proteins from distal proteins, which are away from the core PB machinery. Based on the LFQ values of all biotinylated peptides, we quantified the true biotinylated proteins. Throughout the statistical analysis of quantified biotinylated protein abundance, differentially enriched proteins were classified by comparing each PB-targeted bait to the cytosolic APEX2-NES or TurboID-NES (Fig. 6c; Supplementary Data 5). Based on this result, we classified 158 and 131 proteins in the PB proteome that were enriched in PB-targeted experiments using APEX2 and TurboID, respectively (Fig. 6d). Our proteomic analysis of PB proteome profiling based on two different proximity labeling enzymes showed a number of overlapping lists of proteins. This result was also compared to a published PB proteome[31] that used BioID conjugated to the same bait proteins as those in our experiment (Fig. 6a), but that used a conventional ratiometric approach to analyze enriched PB candidates. This comparative analysis evaluated overlapping proteins between experiments and compared the number of identified PB candidates. Consequently, the core PB proteins were well-identified using different baits and conjugated proximity labeling enzyme combinations, and the number of exclusively identified proteins related to PB from different baits was also defined (Fig. 6d). To evaluate the biotinylation site identification results of APEX2 and TurboID experiments, network analysis of 258 proteins listed as PB candidates was carried out, which revealed that enriched PB core components showed dense networks with other identified biotinylated proteins, evidencing biological interactions (Fig. 6e). GO enrichment analysis of captured biotinylated PB proteins corresponded with the expected results and enhanced GO terms of RNP granules, post-transcriptional regulation of gene expression, and post-transcriptional gene-silencing terms (Fig. 6f).

Through this process, we identified a list of proteins that were highly enriched without PB annotation. UBAP2L, a protein originally found only in stress granules (SG)[31], was labeled in APEX2 and TurboID experiments. To validate this finding and confirm the PB localization of UBAP2L, eGFP-tagged UBAP2L was transfected into the HEK293t-cell line, and immunofluorescence analysis was carried out using confocal microscopy under various cellular stress conditions. Our results confirmed that UBAP2L was not only present in PB under normal conditions, but it could also localize under oxidative stress conditions. Additionally, under arsenite-treated stress conditions, UBAP2L showed non-PB-localized granules, which were suspected to be SG. This observation strongly suggests that UBAP2L has dual localization in PB and SG (Fig. 6g).

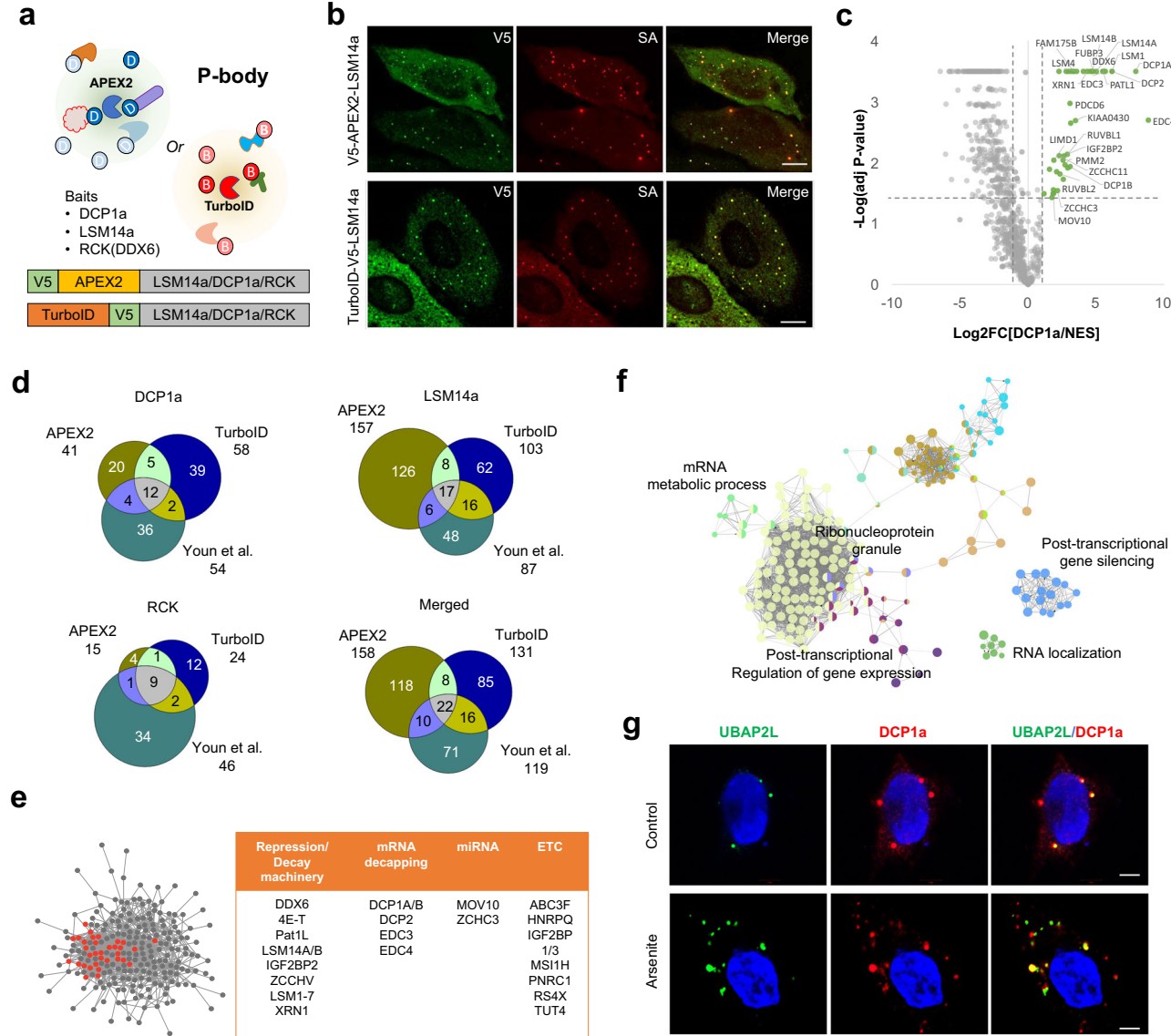

**Fig. 6 | Application of super-resolution proximity labeling to map p-body proteome. a** Schematic illustration of processing body (PB) proteome profiling via proximity labeling and a list of bait with schematics of bait-APEX2/TurboID construct. **b** Representative immunofluorescence (IF) images of expressed APEX2/TurboID conjugated baits and biotin-labeled patterns. Scale bar = 10 μm. **c** Volcano plot of differentially expressed proteins identified from DCP1a-APEX2 and NES-APEX2 construct. PB candidate proteins are marked green, and gene names of known PB proteins are labeled. **d** Identification overlaps of identified PB candidates among different bait-APEX2/TurboID constructs and published results based on BioID. **e** Network analysis of identified PB protein candidates. Listed core PB proteins (right) are marked red. **f** Gene Ontology analysis of identified PB protein candidates and their network using Clue-go analysis. **g** Representative IF images of UBAP2L localization to PB under various conditions. Arsenite was treated for 30 min. Scale bar = 5 μm.

## Discussion

In this study, we demonstrated a newly developed streptavidin bead-based enrichment method for biotinylated peptides that are highly compatible with LC–MS for identifying the precise location of a biotinylation site via proximity labeling, showing greater depth than related approaches. Although the anti-biotin Ab approach is a solid protocol that is biologically friendly and simple, our novel method combined with the DBP probe demonstrated excellent results in terms of identification and enrichment efficiency. These results may be owing to several drawbacks of Ab capture systems that use BP probes. Additionally, we previously defined the disadvantages of BP probes, including the fact that 5–20% can be oxidized[17], leading to increased sample complexity for LC–MS analysis, which adversely affects the identification of various distinct peptide species. Another possible drawback of the AbB capture approach is the presence of the antibody itself. In contrast, streptavidin shows a much higher affinity for

biotin with stable structural integrity, even under extreme conditions, such as high heat/salt conditions[32–34]. These advantages of streptavidin for washing non-specifically bound peptides under extremely stringent conditions without disrupting the streptavidin-biotin-binding environment cannot be achieved using Ab-based capturing approaches. Thus, inferior enrichment efficiency of antibody-based capture methods is inevitable under mild washing conditions, which reduces the chances of correctly identifying the labeled peptide through LC–MS analysis. The conditions evaluated include the performance of streptavidin or neutravidin from different manufacturers, which showed no significant difference in enrichment (Fig. S1a–c, Supplementary Data 6). Through such improvements, we developed an efficient and precise method for identifying biotinylated peptides and detecting biotin modifications that have been in situ-labeled via proximity labeling.

Using our method, we were able to successfully profile the proteome of the membranous organelles of the mitochondrial matrix, IMS, and PB. Systematic determination of the subcellular/sub-organellar localization of each protein was successfully carried out. The main advantage of identifying the biotinylation site in this process is that it is easy to define FP proteins resulting from non-specific bead binding and proteins that interact with biotinylated proteins, which are difficult to classify using conventional methods. Thus, we reliably obtained a precise list of mitochondria-resident proteomes and PB-associated proteins, including novel candidates.

Moreover, a marked advantage of knowing the exact biotinylation sites obtained by labeling each side of a membrane-bound organelle is that topology mapping of transmembrane proteins can be carried out successfully. Generally, mapping the topology of membrane proteins is based on defining the protein structure through crystallography or biochemical experiments, such as reporter systems. The major limitation of these conventional methods is that systematic and proteome-wide mapping is limited and restricted to a single target protein per experiment. Conversely, proximity labeling coupled with our precise biotinylation site identification method can easily overcome the limitations of biochemical experiments and provide a precise, systematic, and proteome-wide mapping of transmembrane proteins present in vivo. However, under several circumstances, biotinylated protein identification and topology determination by proximity labeling might also be restricted, as in the case of failure to biotinylated lysine or tyrosine residues that are not available for labeling, owing to extensive PTM or limited surface exposure resulting from structural considerations. Additionally, even though our new approach for identifying biotinylation sites facilitated advanced identification and precise workflow (Fig. 3), multiple aspects of the experiment should be carefully considered when applying or comparing it to different biological systems or techniques. Proximity labeling experiments coupled with mass spectrometry are influenced by various factors. These factors include not only experimental variables, such as the expression of proximity labeling enzyme-tagged bait proteins and cell type but also technical considerations. Technical aspects involve the initial cell quantity, biotin-labeling conditions, and optimization of protocol steps relevant to the cellular labeling context. Examples of such technical considerations include determining the amount of streptavidin bead usage and selecting the appropriate bead type for pulling down biotinylated peptides.

The growing significance of MS-based identification of biotinylated proteins and their biotinylation sites is underscored by the advancements in proximity-dependent biotinylation. Our findings support the biotinylation site identification method as an efficient and reliable tool for the characterization of labeled proteins using promiscuous enzymes with wide application potential. To perform precise biological experiments, we expect our methodology to enhance the coverage and identification of biotinylated proteins via proximity labeling in situ.

## Methods

### Plasmids and cloning

Genes were cloned into the specified vectors using standard enzymatic restriction digest and ligation with T4 DNA ligase. To generate constructs where short tags (e.g., V5 or Flag epitope tag) or signal sequences were appended to the protein, the tag was included in the primers used to PCR-amplify the gene. PCR products were digested with restriction enzymes and ligated into cut vectors (e.g., pcDNA3 and pcDNA5). In all cases, the CMV promoter was used for expression in mammalian cells. The genetic constructs cloned and used for this study are summarized in Supplementary Table 1.

### Cell culture and transfections

HEK293 was obtained from ATCC (Manassas, VA, USA) and HEK293 Flip-in T-rex was obtained from Thermo Fisher Scientific (Cat. No. R78007). The cell lines were frequently checked and tested for morphology under a microscope. Cells were maintained in a high glucose DMEM medium with 10% fetal bovine serum at 37 °C in 5% CO2 (v/v).

### Construction of stably expressed cell line and culture (Flip-in HEK293T-Rex)

Flip-In™ T-REx™ 293 cells (Life Technologies) were cultured in DMEM (Hyclone) supplemented with 10% FBS, 2 mM L-glutamine, 50 units/mL penicillin, and 50 μg/ml streptomycin at 37 °C under 5% CO$_2$. Cells were grown in a T25 flask. Stable cell lines were generated depending on the expression system, such as pcDNA™3 expression construct for the constant expression system and pcDNA™5/FRT/TO expression construct for the doxycycline-inducible system. In the case of pcDNA™3 expression construct for constant expression system, the expression constructs were transfected at 60–80% confluence using TurboFect (Invitrogen), typically with 6 μL of TurboFect transfection reagent and 2000 ng plasmid per six-well cell culture plate. After 24 h, cells were split into a 90 mm cell culture dish (SPL, 11090) with the proper concentration of Geneticin (G418) (500 μg/mL). Media containing Geneticin (G418) were changed every 3–4 days. After 2–3 weeks, 3–4 colonies were selected and transferred to a 24-well plate. Cells were continuously split into larger plates, and a cell stock was made. After splitting the cells into a six-well plate, separate samples were prepared for expression testing.

For doxycycline-inducible stable cell line, pcDNA™5/FRT/TO expression construct and the pOG44 plasmid were transfected using TurboFect (Invitrogen), typically with 12 μL TurboFect and 4000 ng plasmid (9:1 = pOG44:pcDNA5) per T25 flask. After 24 h, cells were split into a 90 mm cell culture dish (SPL, 11090) with the proper concentration of hygromycin B (100 μg/mL). Hygromycin B-containing media was changed every 3–4 days. After 2–3 weeks, 3–4 colonies were selected and transferred to a 24-well plate. Cells were continuously split into larger plates, and after splitting the cells into a six-well plate, separate samples were prepared for expression testing. Unlike constant stably expressing cell lines, doxycycline-inducible stable cell lines were induced by 5 ng/mL doxycycline (Sigma Aldrich) for inducible expression.

### Desthiobiotin-phenol (and biotin-phenol) labeling in stably expressed APEX cell line

Stable cells were grown in four T75 flasks. Cells were induced with doxycycline at 60–80% confluence. After 18–24 h, the medium in each flask was changed to 7.5 mL of fresh growth medium containing 250 μM desthiobiotin-phenol or biotin-phenol. All DBP and BP parallel experiments were performed simultaneously with the same protocol. The flasks were incubated at 37 °C under 5% CO$_2$ for 30 min according to previously published protocols. Afterward, 750 μL of 10 mM H$_2$O$_2$ (diluted from 30% H$_2$O$_2$, Sigma Aldrich H1009) was added to each flask for a final concentration of 1 mM H$_2$O$_2$, and the flasks were gently agitated for 1 min at room temperature. The reaction was then quenched by adding 7.5 mL of DPBS containing 10 mM Trolox, 20 mM sodium azide, and 20 mM sodium ascorbate to each flask. Then, the solution was removed, and the cells were washed three times with a cold quenching solution (DPBS containing 5 mM Trolox, 10 mM sodium azide, and 10 mM sodium ascorbate). Cells were detached using of cold quenching solution and centrifuged at 1,500×*g* for 5 min at 4 °C. Cells were resuspended with fresh cold quenching solution and centrifuged again.

### Immunofluorescence and confocal microscopy

To visualize the subcellular localization of the transiently expressing POI, cells were plated on coverslips (thickness no. 1.5 and radius: 18 mm). For fixed cell imaging, cells were fixed with 4% paraformaldehyde and permeabilized with cold methanol for 5 min at −20 °C. Next, cells were washed with Dulbecco's phosphate-buffered saline (DPBS) and blocked for 1 h with 2% BSA in DPBS at room temperature. To detect APEX2-fusion expression, cells were incubated with mouse anti-V5 antibody (Invitrogen, cat. no. R960-25, 1:5000 dilution) for 1 h at room temperature. After washing four times with TBST each 5 min, cells were simultaneously incubated with secondary Alexa Fluor 488-goat anti-mouse IgG (Invitrogen, cat. no. A-11001, 1:1000 dilution) and streptavidin–Alexa Fluor 568 IgG (Invitrogen, cat. no. S11226, 1:1000 dilution) for 30 min at room temperature. Cells were

then washed four times with TBST each for 5 min. Immunofluorescence images were obtained and analyzed SP8 X, Leica (NICEM in Seoul National University, Korea) with objective lens (HC PL APO ×100/1.40 OIL), White Light Laser (WLL, 470–670 nm, 1 nm tunable laser), HyD detector, and controlled by LAS X software.

### Enrichment of biotinylated protein (conventional approach)

Enrichment of biotinylated protein basically followed the experimental procedure of Cho et al. (2020)[1]. Briefly, cell pellets were lysed with RIPA lysis buffer (50 mM Tris, 150 mM NaCl, 0.1% SDS, 0.5% sodium deoxycholate, 1% Triton X-100), 1× protease cocktail (Sigma Aldrich, P8849), 1 mM PMSF (phenylmethylsulfonyl fluoride), 10 mM sodium azide, 10 mM sodium ascorbate, and 5 mM Trolox for 10 min at 4 °C. Lysates were clarified by centrifugation at 13,000×$g$ for 10 min at 4 °C. prewashed streptavidin magnetic bead was incubated with sample for at least 1 h with end-to-end rotation. After enrichment, beads were washed twice with RIPA buffer, once with 50 mM Tris–HCl (pH 7.5), followed by two washes with 2 M urea in 50 mM Tris–HCl (pH 7.5) buffer, then incubated with fresh 2 M Urea in 50 mM Tris–HCl (pH 7.5) containing 1 mM DTT and trypsin at 25 °C for 1 h. then streptavidin beads were washed twice with 2 M urea in 50 mM Tris buffer (all the supernatant was combined). Reduction of disulfide bond and alkylation was carried out, then samples were digested with trypsin overnight at 37 °C. After digestion, samples were acidified with formic acid and desalted for LC–MS analysis.

### Enrichment of biotinylated peptides

Harvested samples were lysed with 2% SDS in 1× TBS (25 mM Tris, 0.15 M NaCl, pH 7.2, Thermoscientific, 28358), 1× protease inhibitor cocktail. Lysates were clarified by ultrasonication (Bioruptor, diagenode) for 15 min with the cold water bath. To remove free probes, six times the sample volume of cold acetone (−20 °C, Sigma-Aldrich, 650501) was added to each lysate and kept at −20 °C. After at least 2 h, samples were centrifuged at 15,000×$g$ for 15 min at 4 °C. Supernatant was gently removed, and acetone precipitation was repeated with 5 mL of cold acetone containing 1/6 volume of 1× TBS. After removing the supernatant, the pellet was solubilized with 8 M urea (Sigma-Aldrich, U5378) in 50 mM ammonium bicarbonate (ABC, Sigma-Aldrich, A6141). The concentration of protein was measured, and samples were denatured at 650 rpm for 1 h at 37 °C. After denaturation, the samples were reduced with the final concentration of 10 mM dithiothreitol (Sigma-Aldrich, 43816) and incubated at 650 rpm for 1 h at 37 °C. The samples were alkylated by adding iodoacetamide (Sigma-Aldrich, I1149) of 40 mM to the final concentration and mixed at 650 rpm for 1 h at 37 °C. The samples were diluted eight times with 50 mM ABC, and CaCl$_2$ (Alfa aesar, 12312) was added to achieve a 1 mM final concentration. ~2 mg of protein sample was digested by adding trypsin (Thermoscientific, 20233, 50:1 w/w) and incubated at 650 rpm for 6–18 h at 37 °C. After trypsinization, digested peptide samples were incubated with 150 μL of streptavidin beads (Pierce, 88817), which were pre-washed with 2 M urea in 1× TBS four times prior to sample enrichment. The samples were incubated for 1 h at room temperature with end-over-end rotation. To remove nonspecific bound peptides, the beads were washed twice with 2 M urea in 50 mM ABC and then finally washed with distilled water. To eluted biotinylated peptides, elution buffer [80% acetonitrile (Sigma-Aldrich, 900667), 0.2% TFA (Sigma-Aldrich, T6508), and 0.1% formic acid (Thermoscientific, 28905)] was added and incubated at 60 °C for 5 min. Each supernatant was transferred to new tubes, and the elution step was repeated for 2–3 times. Combined elution fractions were dried using Speed vac (Eppendorf). Samples can be stored at −20 °C or injected into mass spectrometry directly.

### Enrichment of biotinylated peptides (Spot-ID)

Cells were lysed with RIPA lysis buffer (50 mM Tris, 150 mM NaCl, 0.1% SDS, 0.5% sodium deoxycholate, 1% Triton X-100), 1× protease cocktail (Sigma Aldrich, P8849), 1 mM phenylmethylsulfonyl fluoride (PMSF), 10 mM sodium azide, 10 mM sodium ascorbate, and 5 mM Trolox for 10 min at 4 °C. Lysates were clarified by centrifugation at 15,000×$g$ for

20 min at 4 °C. For removal of unreacted free desthiobiotin-phenol or biotin-phenol, cell lysates were moved into Amicon filter (Merck Millipore, 10 kDa-off) followed by centrifugation at 7500×$g$ for 15 min at 4 °C. Phosphate-buffered saline (PBS) containing 1 mM PMSF and 1× protease cocktail was added up to 4 mL followed by centrifugation three more times. Finally, the cell lysates were transferred to an Eppendorf tube and mixed with 300 μL of streptavidin beads (Pierce). The sample was rotated for 1 h at room temperature and washed twice with PBS. After removing the PBS, 100 μL of denaturing solution (6 M urea, 2 M thiourea, 10 mM HEPES) was added and reduced using 20 μL of 100 mM DTT (dithiothreitol) in 50 mM ammonium bicarbonate (ABC) buffer for 60 min at 56 °C using a Thermomixer (Eppendorf). Protein alkylation was performed by adding 35 μL of 300 mM iodoacetamide in 50 mM ABC buffer with shaking in the dark for 30 min. Afterward, trypsin gold (Promega) was added to the solution and incubated at 37 °C with shaking overnight. Afterward, formic acid was added to terminate the trypsin reaction, and the beads were washed with PBS four times and eluted by boiling at 95 °C for 10 min after adding 250 μL of 95% formamide, 10 mM EDTA, pH 8.2. Eluted peptide samples were desalted with Varian Bond ELUT (Agilent, 12109301) and homemade column.

### Desalting enriched biotinylated peptide (Spot-ID)

Varian Bond ELUT was activated with 1 mL 3% acetonitrile/0.1% formic acid, 1 mL 100% acetonitrile, and 2 mL 3% acetonitrile/0.1% formic acid (v/v). Then, the sample was added to the column for binding (×2). The column was washed using 2 mL 3% acetonitrile/0.1% formic acid (v/v). It was then eluted with 1.4 mL of 70% acetonitrile/0.1% formic acid (v/v) and 500 μL of 100% acetonitrile. The eluted fraction was dried. Dried samples were redissolved in the 0.1% formic acid. A home-made column was used for obtaining more clean samples. The end of a 200-μL Eppendorf tip was blocked with a 3 M Empore C8 disk (3 M Bioanalytical Technologies, 2214) and ~5 mg of POROS Oigo R2 reversed phase resin (Applied Biosystems, 1-1159-06) was added. The column was—S7-activated by sequential centrifugation at 1000×$g$ with 100 μL of 3% acetonitrile/0.1% formic acid (v/v), 100 μL of 100% acetonitrile, and 200 μL of 3% acetonitrile/0.1% formic acid (v/v). Then, the samples in the 0.1% formic acid were added to the column. The column was washed using 200 μL of 3% acetonitrile/0.1% formic acid (v/v) three times. The column was then eluted with 200 μL of 70% acetonitrile/0.1% formic acid (v/v) twice and then with 50 μL of 100% acetonitrile. The eluted fraction was dried in a speed vac and kept in the refrigerator.

### Enrichment of biotinylated peptides (anti-biotin antibody)

Cells were lysed with 8 M urea lysis buffer, 50 mM ammonium bicarbonate (ABC), 1× protease cocktail (Sigma Aldrich, P8849), 1 mM phenylmethylsulfonyl fluoride (PMSF), 10 mM sodium azide, 10 mM sodium ascorbate, and 5 mM Trolox for 10 min at 4 °C. Lysates were clarified by centrifugation at 15,000×$g$ for 20 min at 4 °C. proteins were reduced with 10 mM DTT for 1 h at 37 °C and subsequently carbaminomethylated with 40 mM Iodoacetamide for 1 h at 37 °C in the dark. Then the urea concentration was reduced to 1 M with 50 mM ABC. Samples were digested overnight at 37 °C with trypsin (Thermoscientific, 20233, 50:1 w/w). following digestion, samples were acidified with formic acid and desalted on a C18 SPE cartridge (SUPELCO). Dried samples were dissolved in 50 mM MOPS (pH 7.2), 10 mM sodium phosphate, and 50 mM NaCl, then incubated with anti-biotin antibody (ImmuneChem Pharmaceuticals Inc, ICP0615). Samples were incubated with end-over-end rotation for 1 h at 4 °C. The solutions were spun down at 1000×$g$ for 1 min. The flowthrough was removed. Antibody beads were washed 4× with 1.5 ml of ice-cold PBS. Biotinylated peptides were eluted from the antibody with 50 μl of 0.15% TFA. Antibody beads were spun down at 1000×$g$ for 1 min, and the eluent was transferred to a new microfuge tube. This process was repeated, the eluents were combined, and the samples were desalted.

**Article**

## Elution buffer comparison

Biotinylated peptides, which are labeled with DBP, were enriched with streptavidin as described in the above section then samples were either eluted with Laemmli buffer heated for 10 min at 60 °C or acidic organo-aqueous elution buffer [80% acetonitrile (Sigma-Aldrich, 900667), 0.2% TFA (Sigma-Aldrich, T6508), and 0.1% formic acid (Thermoscientific, 28905)] incubated at 60 °C for 5 min. For dot blot assay, the sample eluted with Laemmli buffer was diluted with TBS to reduce SDS concentration. An equal amount of diluted Laemmli buffer was added into the sample eluted with elution buffer after drying acetonitrile in a speed-vac. Subsequently, a dot-blot assay (Biorad) was carried out using a nitrocellulose membrane with a 0.2 μm pore size. Samples were applied to the wells, allowing the entire sample to pass through the membrane via gravity flow. The membrane was allowed to air-dry at room temperature for several minutes to evaporate a liquid, followed by rinsing with TBS. Immunoassay was conducted using streptavidin-HRP (Thermoscientific, 21124). The intensity of the blot was measured using ImageJ.

## LC–MS/MS analysis of enriched peptide samples

Analytical capillary columns (100 cm × 75 μm i.d.) and trap columns (2 cm × 150 μm i.d.) were packed in-house with 3 μm Jupiter C18 particles (Phenomenex, Torrance). The long analytical column was placed in a column heater (Analytical Sales and Services) regulated to a temperature of 45 °C. NanoAcquity UPLC system (Waters, Milford) was operated at a flow rate of 300 nL/min over 2 h with a linear gradient ranging from 95% solvent A ($H_2O$ with 0.1% formic acid) to 40% of solvent B (acetonitrile with 0.1% formic acid). The enriched samples were analyzed on an Orbitrap Fusion Lumos mass spectrometer (Thermo Scientific) equipped with an in-house customized nanoelectrospray ion source. Precursor ions were acquired ($m/z$ 300–1500) at 120 K resolving power and the isolation of precursor for MS/MS analysis was performed with a 1.4 Th. Higher-energy collisional dissociation (HCD) with 30% collision energy was used for sequencing with an auto gain control (AGC) target of 1e5. Resolving power for acquired MS2 spectra was set to 30k at with 200 ms maximum injection time.

## MS data processing and protein identification

All MS/MS data were searched by MaxQuant (version 1.6.2.3) with Andromeda search engine at 10 ppm precursor ion mass tolerance against the SwissProt Homo sapiens proteome database (20,199 entries, UniProt (http://www.uniprot.org/), 2017-08-06). The label-free quantification (LFQ) and match between runs were used with the following search parameters for the TurboID experiment: semi-trypic digestion, fixed carbaminomethylation on cysteine, dynamic oxidation of methionine, protein N-terminal acetylation with biotin (+226.0775) labels of lysine residue. The following parameters were used for the APEX2 experiment: tryptic digestion, fixed carbaminomethylation on cysteine, dynamic oxidation of methionine, protein N-terminal acetylation with either BP (+361.146) or DBP (+331.1896) of tyrosine residue, based on its usage. Less than 1% of false discovery rate (FDR) was obtained for unique labeled peptides and as well as unique labeled proteins. LFQ intensity values were log-transformed for further analysis and missing values were filled by imputed values representing a normal distribution around the detection limit. To impute the missing value, first, the intensity distribution of mean and standard deviation was determined, then for imputation values, a new distribution based on Gaussian distribution with a downshift of 1.8 and width of 0.3 standard deviations was created for the total matrix.

## Cluster analysis and gene ontology enrichment analysis

By using Perseus (Ver1.6.2.3), each set of experiments is normalized scale to the interval (0–1) independently and subject to hierarchical clustering analysis with the following option: Euclidean distance with option of preprocessed with $k$-means clustering, 10 maximal number of iterations and preserving the order of column constraint. Defined genes with each cluster were subjected to gene ontology enrichment analysis using the DAVID online tool (https://david.ncifcrf.gov/), the background used for

Go analysis for each cluster was set as the entire proteome of *Homo Sapiens*

## Statistics and reproducibility

All the statistical tests and informatic analysis with data visualization were performed using Perseus (v.1.6.2.3) with implemented WGCNA R-package, GraphPad Prism software (v10.1.2), InfernoRDN, Orange, R studio, and Microsoft Excel. Data are shown as mean ± standard deviation (SD). Comparison between groups was done through unpaired two-tailed Student's *t*-test. All the experiments were done in triplicate unless indicated.

## Reporting summary

Further information on research design is available in the Nature Portfolio Reporting Summary linked to this article.

## Data availability

The mass spectrometry proteomics datasets have been deposited to the ProteomeXchange Consortium via the PRIDE[35] partner repository with the dataset identifier PXD047979. All processed results are provided as Supplementary Data.

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

## Acknowledgements

This work was supported by the National Research Foundation of Korea (Grant nos. NRF-2021R1A2C2009336, NRF-2019M3E5D3073104, NRF-2022M3A9I2082294 to J.-S.K.; NRF-2022R1A2B5B03001658 to H.-W.R.; NRF-2021R1A6A3A01087055 to S.S.; NRF-2020R1C1C1013927 to M.G.K.), Comparative medicine Disease Research Center (NRF-2021R1A5A1033157), and Institute for Basic Science from the Ministry of Science and ICT of Korea (IBS-R008-D1 to J.-S.K.). The indicated graphics of the petri dish in Figs. 1 and 2a are created with BioRender.com.

## Author contributions

S.S., S.-Y.L., H.-W.R., and J.-S.K conceived the study. S.S. and S.-Y.L. developed the methodology with the help of M.-G.K. and D.G.J. S.S. and J.K. performed the experiments and data analysis. S.S., S.-Y.L., H.-W.R., and J.-S.K. contributed to the experimental design and data interpretation. S.S., H.-W.R., and J.-S.K. wrote the manuscript with input from all the authors.

## Competing interests

The authors declare no competing interests.
