## [Peer Review File · Communications Biology]

Reviewers' comments:

Reviewer #1 (Remarks to the Author):

This manuscript by Shin et al. describes the application of a method to elute biotinylated peptides off affinity -capture beads for their identification by MS in proximity labeling experiments. The novelty lies in the specific chemical elution of the biotinylated peptides, switching from a previously published formamide from avidin/streptavidin, or the use of anti-biotin antibodies by other groups, to an acetonitrile/TFA/formic acid elution solution, used in combination with previously utilized protein trypsinization prior to affinity capture. This approach was compared with other approaches and shown to yield more expected candidates and detection of biotinylated proteins. Overall, the methodical advance in the manuscript, which is the centerpiece as presented, is a modest technical improvement, but one that likely may be utilized by many in the field, at least the buffer to elute biotinylated protein/peptides from avidin/streptavidin. The comparative applications support the improvement in the approach, although critical a comparison was not performed. There is likely limited value in the biological knowledge gained from these comparative proximity labeling studies, although that did not appear to be their intent.

Specific comments:

Line 45 replace the word expanded. Perhaps faster or enhanced

The concept in lines 54-61 would only apply to the extremely robust labeling approaches of APEX or perhaps TurboID, not BioID. The downsides of APEX or TurboID, at least in terms of non-specific labeling, are not that irrelevant proteins are detected due to background during the affinity capture, it is that irrelevant proteins are labeled in the cells due to excessive activity of those enzymes. No downstream post-labeling method can correct that, outside of use of proper controls and sophisticated quantitative MS approaches.

Please include methods for all experiment performed. I did not see anti-biotin enrichment or Spot-ID techniques described.

Line 72, please cite the previous study.

Line 93-94, "...followed by lysis of proximity-labeled cells." Makes no sense as you have just described taking a protein lysate and precipitating it, thus the cells have already been lysed.

The only major novel aspect of this manuscript is the use of a poorly defined "acidic organo-aqueous denaturation buffer" line 95-96. This information is found buried in the supplemental file (where all of the methods oddly are) and consists of 80% acetonitrile, 0.2% TFA and 0.1% formic acid. It would be preferable to discuss in the results the details of this elution buffer that lies at the heart of this manuscript. It should also be shown that this elution method is superior to alternate ones, either in the elution of biotinylated protein and/or the reduction of contaminating streptavidin.

It is unclear to me if an experiment was done using the newly described biotin-elution protocol on conventionally purified proteins (i.e. intact proteins), as is typically done in a proximity labeling experiment. I appreciate that there may a loss of sensitivity in detecting biotinylated peptides as they will no longer be enriched, but to what extent? There are downsides to only purifying biotinylated peptides, most notably that not all peptides are equally detectable by MS, but an intact protein is

more likely to provide peptides that are detectable. Presumably this new elution approach may solve much of the downsides of on-bead digestion, most notably the considerable abundance of streptavidin peptides, but what other background, if any, does it prevent? This experiment needs to be done and highlighted to reveal the true potential of this advance in biotin-elution approaches.

The use of controls (line 215-216) goes against one of the claimed advantages of this new protocol.

In fig 3b, why does Spot-ID require 10mg and this study only requires 2.5mg? Can the authors please be more clear about this?

Any background from any of these methods should be easily removed by requiring the biotin (or biotin-like) modification to be calculated in the mass of the purified and MS-identified peptides.

Please cite relevant papers of similar methods: e.g. PMID: 29249144

Comparisons of detection with other published proximity labeling studies should be done carefully if there is a claim of detection of more relevant proteins. This could be the result of various technical issues, including expression of proximity labeling enzymes, labeling conditions, cell types, cell numbers lysed, affinity capture matrix abundance relative to labeled peptides, and MS techniques and analysis thresholds. Even if performed by the same investigators, these factors can play a substantial role in differences and should be noted.

The graphs in Figure 4(b,c,d,e) appear to be the same graphs in S2(a,b,d,h) - why show them twice?

MS results need to be provided for all experiments as Excel files.

Reviewer #2 (Remarks to the Author):

Shin et al. described a method for enriching biotinylated peptides derived from proximity-dependent biotinylation by combining desthiobiotin phenol, streptavidin beads, and acid elution. Using the method, they successfully identified thousands of biotinylated peptides with high specificity (~89%). Finally, Shin et al. applied the method to study the processing body (PB) and identified novel candidates such as UBAP2L.

Overall, the manuscript is well-written, and designed, although similar works about the enrichment of biotinylated peptides using avidin-relatives were reported.

1. Line 95: It is not clear why biotinylated peptides can be eluted from streptavidin beads using an acidic solution. Neuroavidin should be possible due to lower stability. It would be more helpful for readers to explain why acid elution is possible.

2. Figure 3b: This time why did the authors use avidin but not streptavidin?

3. Line 149: Please provide evidence that DBP is more efficient than BP for LC/MS analysis (before reaching Figure 4).

4. The authors should cite, compare, and discuss the relevant papers where avidin-relatives were used to enrich biotinylated peptides;

Schiapparelli, L.M., McClatchy, D.B., Liu, H.-H., Sharma, P., Yates, J.R., 3rd, and Cline, H.T. (2014) Direct detection of biotinylated proteins by mass spectrometry. *J. Proteome Res.* 13, 3966–3978

Prikas, E., Poljak, A., and Ittner, A. (2020) Mapping p38 α mitogen-activated protein kinase signaling by proximity-dependent labeling. *Protein Sci.* 29, 1196–1210

Kohei Nishino, Harunori Yoshikawa, Kou Motani, and Hidetaka Kosako (2022) Optimized Workflow for Enrichment and Identification of Biotinylated Peptides Using Tamavidin 2-REV for BioID and Cell Surface Proteomics. *J. Proteome Res.*

Method part5. 150 μ L of streptavidin beads sounds like a large volume to me. Is this for one experiment?

6. Which version (year/month) of the SwissProt database was used?

7. Please add information about the mass of the (desthio)biotin label of a lysine residue.

Reviewer #3 (Remarks to the Author):

Comments to the Author

Systematical mapping of spatial protein localization and protein-protein interactions in cells is the central task of cell biology. These data provide valuable information about almost all vital processes in cells, such as DNA replication, gene transcription, and translation, cell-cycle control and proliferation, signal transduction, and cell-cell communication. Proximity labeling techniques such as APEX and Turbo-ID, based on using engineered ascorbate peroxidase and biotin ligase mutants are now widely used for the study of intracellular phenomena. The authors Sanghee Shin et al present a work where they applied a super-resolution proximity labeling method with enhanced direct identification of biotinylation sites.

This work is a continuation of the previously published authors' method Spot-ID using desthiobiotin-phenol1 but also original with some improvements and simplification of experimental design. The authors developed a new super-resolution proximity labeling method that directly reveals the biotinylation sites based on peptide-level enrichment. In comparison with the conventional ratiometric approach utilizing protein level enrichment, the new method outperformed in terms of simpler experimental design, data reliability, LC-MS utility, and mostly unbiased identification of biotinylated proteins. The manuscript does not have technical or conceptual flaws which prohibits its publication.

I would recommend this review for publication in *Communications Biology* with some edits, which I feel will improve the clarity of the work for the broader readership.

1. Did the authors check the difference in elution efficiency of DBP and BP-modified proteins and peptides from the SA-beads using an acidic organo-aqueous denaturation buffer? If it is a known elution method, please add a citation. Did the authors compare the efficiency of this elution with traditional elution using boiling in Laemmli buffer with 1-3% of SDS², 3 on protein (Western blotting) or peptide (PRM/MRM) level? (Line 95 “The enriched biotinylated peptides were eluted using acidic

organo-aqueous denaturation buffer”; In section Enrichment of biotinylated peptides (Supplementary information) "To eluted biotinylated peptides, elution buffer [80% acetonitrile (Sigma-Aldrich, 900667), 0.2% TFA (Sigma-Aldrich, T6508), and 0.1% formic acid (Thermoscientific, 28905)] was added and incubated at 60°C for 5 min."

2. The authors studied the topology of the mitochondrial matrix, inner membrane, and intermembrane space proteome via proximity labeling using APEX2 construct. But did they also confirm these data by using MTS-Turbo-ID and SCO1-Turbo-ID? (Line 281. "However, under several circumstances, topology determination by proximity labeling might also be restricted, as in the case of failure to biotinylate lysine or tyrosine residues that are not available for labeling, owing to extensive PTM or limited surface exposure resulting from structural considerations.")

3. Please submit your LC-MS/MS raw data to proteome repository Pride, Peptide Atlas, or other⁴ and include accession number in data availability section of the manuscript.

4. Please add the meaning of the abbreviation and a brief explanation of the choice of this peptide sequence MTS. (Line 107. MTS-APEX ->24-amino-acid mitochondrial targeting sequence (MTS) from COX4 to localize APEX2 throughout the entire mitochondrial matrix⁵.)

5. Line 110. NC negative control

6. Please add legend to lower figure topology mapping as "This paper" in Fig. 5e.

REFERENCES

1. Lee, S.Y. et al. Architecture Mapping of the Inner Mitochondrial Membrane Proteome by Chemical Tools in Live Cells. *J Am Chem Soc* 139, 3651-3662 (2017).
2. de Boer, E. et al. Efficient biotinylation and single-step purification of tagged transcription factors in mammalian cells and transgenic mice. *Proc Natl Acad Sci U S A* 100, 7480-7485 (2003).
3. Villasenor, R. et al. ChromID identifies the protein interactome at chromatin marks. *Nat Biotechnol* 38, 728-736 (2020).
4. Deutsch, E.W. et al. The ProteomeXchange consortium in 2020: enabling 'big data' approaches in proteomics. *Nucleic Acids Res* 48, D1145-D1152 (2020).
5. Han, S. et al. Proximity Biotinylation as a Method for Mapping Proteins Associated with mtDNA in Living Cells. *Cell Chem Biol* 24, 404-414 (2017).

Reviewers' comments:

Reviewer #1 (Remarks to the Author):

This manuscript by Shin et al. describes the application of a method to elute biotinylated peptides off affinity -capture beads for their identification by MS in proximity labeling experiments. The novelty lies in the specific chemical elution of the biotinylated peptides, switching from a previously published formamide from avidin/streptavidin, or the use of anti-biotin antibodies by other groups, to an acetonitrile/TFA/formic acid elution solution, used in combination with previously utilized protein trypsinization prior to affinity capture. This approach was compared with other approaches and shown to yield more expected candidates and detection of biotinylated proteins. Overall, the methodical advance in the manuscript, which is the centerpiece as presented, is a modest technical improvement, but one that likely may be utilized by many in the field, at least the buffer to elute biotinylated protein/peptides from avidin/streptavidin. The comparative applications support the improvement in the approach, although critical a comparison was not performed. There is likely limited value in the biological knowledge gained from these comparative proximity labeling studies, although that did not appear to be their intent.

Response: We thank the reviewer for finding our work ambitious and comprehensive. We have also fully taken on the major suggestions made by the reviewers, addressed individually below.

Specific comments:

Line 45 replace the word expanded. Perhaps faster or enhanced

Response: The word was modified to enhanced.

The concept in lines 54-61 would only apply to the extremely robust labeling approaches of APEX or perhaps TurboID, not BioID. The downsides of APEX or TurboID, at least in terms of non-specific labeling, are not that irrelevant proteins are detected due to background during the affinity capture, it is that irrelevant proteins are labeled in the cells due to excessive activity of those enzymes. No downstream post-labeling method can correct that, outside of use of proper controls and sophisticated quantitative MS approaches.

Response: We agree with the reviewer that irrelevant proteins are labeled in the cells due to excessive activity of enzymes. We have added additional information regarding the excessive activity for clarity:

“Several factors can contribute to this phenomenon, such as the labeling of non-specific proteins due to excessive enzyme activity or enrichment of secondary (or non-specific) proteins interacting with biotin-labeled proteins (Fig. 1a). Consequently, it is difficult to determine whether the unannotated, enriched proteins are FP or TP using conventional approaches.”

Please include methods for all experiment performed. I did not see anti-biotin enrichment or Spot-ID techniques described.

Response: We have thoroughly revised the Methods section to ensure that we have included all necessary details for the methods we used.

Line 72, please cite the previous study.

Response: The previous study was cited in the result section *“The newly developed method was based on our previous protocol, Spot-ID¹⁷,”*

Line 93-94, “...,followed by lysis of proximity-labeled cells.” Makes no sense as you have just described taking a protein lysate and precipitating it, thus the cells have already been lysed.

Response: These statements have been clarified.

“The protein extract from the lysate was precipitated to remove excess biotin and then resuspended for denaturation with 8 M urea buffer and then digested with trypsin.”

The only major novel aspect of this manuscript is the use of a poorly defined “acidic organo-aqueous denaturation buffer” line 95-96. This information is found buried in the supplemental file (where all of the methods oddly are) and consists of 80% acetonitrile, 0.2% TFA and 0.1% formic acid. It would be preferable to discuss in the results the details of this elution buffer that lies at the heart of this manuscript. It should also be shown that this elution method is superior to alternate ones, either in the elution of biotinylated protein and/or the reduction of contaminating streptavidin.

Response: Figure 2i has been included to illustrate the decrease in streptavidin contamination. Moreover, the results comparing the impact of the elution buffer used in this study with the formamide elution buffer used in Spot-ID have been explained in response to the other comment below (“In fig 3b, why does Spot-ID require 10mg and this study only requires 2.5mg?”). Briefly, as shown in Fig. R2, the acidic organo-aqueous buffer used in this study led to the identification of a significantly higher number of unique biotinylated peptides compared to the formamide elution, as determined through LC-MS analysis.

Figure 4f

Figure 4g

Figure 4. (f). Dot blot results of comparing different elution buffers. (g). Quantitative comparison of dot blot results of eluted samples from Figure 4f and Supplementary Figure 2g; n = 4.

Moreover, despite the non-comparability of peptide elution in the Laemmli buffer (2% SDS) in the LC-MS analysis, we compared elution between the Laemmli (2% SDS) and acidic organo-aqueous buffers (used in this study) using the DBP-labeled peptide (by MTS-APEX2) enriched with streptavidin. As a result, in terms of eluting DBP-labeled peptide from streptavidin, the dot blot analysis showed that the acidic organo-aqueous buffer was significantly superior to the Laemmli buffer (Fig. 4f, g, Supplementary Fig. 2g).

It is unclear to me if an experiment was done using the newly described biotin-elution protocol on conventionally purified proteins (i.e. intact proteins), as is typically done in a proximity labeling experiment. I appreciate that there may be a loss of sensitivity in detecting biotinylated peptides as they will no longer be enriched, but to what extent? There are downsides to only purifying biotinylated peptides, most notably that not all peptides are equally detectable by MS, but an intact protein is more likely to provide peptides that are detectable. Presumably this new elution approach may solve much of the downsides of on-bead digestion, most notably the considerable abundance of streptavidin peptides, but what other background, if any, does it prevent? This experiment needs to be done and highlighted to reveal the true potential of this advance in biotin-elution approaches.

Response: We have clarified our study design. The experiment that we referred to as the “conventional experiment” was compared with our newly developed protocol (see Figure 2). For fair comparison on both experiments regarding the usage of probe, conventional approach used DBP labeling then enrichment/elution was followed the previously published article (Cho et al., 2020). This paper was originally published by the Alice Ting lab, who first demonstrated the proteomic method of identifying biotinylated proteins labeled with APEX2/TurboID.

For clarity we have modified main text as “*We conducted a comparison experiment between the conventional ratiometric approach (Fig. 1a) and newly developed biotinylation site identification method (Fig. 1b) based on the mitochondrial matrix proteome via APEX2 to analyze the efficacy of target protein identification and classification.*” That, each experiment was referred by its own figure of an experimental scheme.

Based on a conventional workflow, we showed 69 biotinylated sites, but only 46 sites showed quantitative values based on the label-free quantification. Among these 46 sites, only six were identified in more than two replicates. Compared to this, our newly developed method identified more than 1300 sites in a single experiment.

We agree with the reviewer that there are several downsides to only purifying biotinylated peptides. Firstly, not all peptides are equally detectable by MS. Based on their nature, certain peptides can be more efficient in ionization, but those that show less efficient ionization can be hard to detect by MS. Secondly, based on the amino acid sequence, tryptic peptide can be either too short or too long, which may be difficult to be detected/sequenced by MS. Finally, as in the case of failure to biotinylate lysine (TurboID) or tyrosine (APEX2) residues that are not available for labeling, owing to extensive PTM or limited surface exposure resulting from structural considerations, we could lose the identification of these proteins.

Regarding the first two points and considering that intact proteins are more likely to provide peptides that are detectable of low abundant proteins, I have profiled the proteome of the HEK293T cell line that we used in our study to map the abundance (intensity-based absolute quantification, iBaq) of each protein. The iBaq value of proteins mapped in Figure 2g was classified into three distinct groups and plotted as a boxplot in Figure R1. Group 1) Mitochondrial matrix proteins that are mapped only in the conventional approach, Group 2) mitochondrial matrix proteins that are mapped in both approaches, and Group 3) mitochondrial matrix proteins that are mapped only in Biotin Site ID.

Figure R1. Protein abundance plot mapped from different approaches.

and plotted as a boxplot in Figure R1. Group 1) Mitochondrial matrix proteins that are mapped only in the conventional approach, Group 2) mitochondrial matrix proteins that are mapped in both approaches, and Group 3) mitochondrial matrix proteins that are mapped only in Biotin Site ID.

According to the analysis presented in Figure R1, 113 mitochondrial matrix proteins exclusively mapped in the conventional approach (intact protein, Group 1) exhibited lower protein abundance compared to that of proteins mapped only using the biotin Site ID method (Group 3). This may indicate that the

conventional approach of enriching intact protein could provide a better chance of detecting peptides from relatively low-abundant proteins compared to the biotin site identification method. However, the degree of difference in the distribution of protein abundance between groups 1 and 3 was marginal. Furthermore, as an alternative method to the conventional workflow, we showed that the current approach not only significantly reduced the streptavidin background (Figure 2i) but also has several other advantages: simpler experimental design, data reliability, LC-MS utility, efficient biotinylated peptide elution (Figure 4f, g), and mostly unbiased identification of biotinylated proteins.

The use of controls (line 215-216) goes against one of the claimed advantages of this new protocol.

Response: As the reviewer mentioned that irrelevant proteins are labeled in the cells due to excessive activity of enzymes, we would reframe the role of APEX2-NES or TurboID-NES to classify core PB proteins and distal PB proteins away from the core structure of PB core machinery. We have changed the sentence to “cytosolic APEX2-NES or TurboID-NES was used as a control to distinguish core PB proteins from distal proteins, which are away from the core PB machinery.”

In fig 3b, why does Spot-ID require 10mg and this study only requires 2.5mg? Can the authors please be more clear about this? Any background from any of these methods should be easily removed by requiring the biotin (or biotin-like) modification to be calculated in the mass of the purified and MS-identified peptides.

Response: Even though Spot-ID showed relatively high enrichment efficiency (72%), the number of biotinylated sites (biotinylated PSM) was significantly lower than that in the other two methods. In Spot-ID (Lee et al., 2017), which was our old protocol, biotinylated proteins are captured with streptavidin beads and then digested by trypsin, while biotinylated proteins are attached on the bead. After trypsinization, unlabeled peptides are washed off and biotinylated peptides are eluted from the streptavidin-bead with **formamide**, which must be **desalted prior to LC-MS analysis**. Additionally, as shown in figure 2i, during the on-bead digestion, streptavidin which holds biotinylated peptides, can also be digested, potentially leading to the loss of captured biotinylated peptide during the washing step. From Spot-ID, we observed that on-bead digestion and **formamide elution** are often inadequate for the capture of biotinylated peptides. As on-bead digestion often results in **insufficient digestion and/or high background noise**, including digested streptavidin may dominate the overall chromatograms, which not only reduces the enrichment efficiency but also disrupts the identification of peptides that are labeled but not as abundant. Furthermore, the use of LC-MS-incompatible formamide buffer for elution requires an additional desalting process that can reduce the yield of biotinylated peptides.

Figure R2. Number of unique biotinylated peptide identified from two different elution buffers. (n=3 for each experiment)

In this study, the biotin-dependent capture by streptavidin-beads was therefore optimized to be carried at the peptide level (**fully digested**), and the formamide elution buffer was replaced with LC-MS-friendly acidic organo-aqueous buffer (80% acetonitrile, 20% water, 0.2% TFA and 0.1% formic acid). The eluted sample was dried without any further sample preparation (which may have led to losses of some of our eluted sample). The resulting sample was directly used for LC-MS analysis only after resuspension with 25 mM ammonium bicarbonate buffer.

To directly assess the impact of the elution buffer, we conducted an additional experiment using samples initiated with 2 mg of mitochondrial matrix labeled with the MTS-APEX2 construct. The process involved our updated method, including (peptide level) streptavidin-based enrichment and washing. Subsequently, two groups of samples were either eluted with the organo-aqueous

denaturation buffer or with formamide, as utilized in our previous Spot-ID study. Then, samples were

analyzed through LC-MS. As shown in Fig. R2, formamide elution identified significantly fewer unique biotinylated peptides compared with the new method. While performing Spot-ID (Lee et al., 2017), we solved this issue of few identifications by increasing the amount of starting sample.

Please cite relevant papers of similar methods: e.g. PMID: 29249144

Response: We cited PMID: 29249144 and several other related papers.

“As separating unlabeled contaminants from the labeled protein subpopulation during LC-MS analysis using the conventional approach is technically demanding, various techniques to identify biotinylation sites from biotin-tagged proteins have been developed that detect biotinylated peptides²⁻⁵.” [Schiapparelli et al., 2014, Kim et al., 2018, Prikas et al., 2020, Nishino et al., 2022]

Comparisons of detection with other published proximity labeling studies should be done carefully if there is a claim of detection of more relevant proteins. This could be the result of various technical issues, including expression of proximity labeling enzymes, labeling conditions, cell types, cell numbers lysed, affinity capture matrix abundance relative to labeled peptides, and MS techniques and analysis thresholds. Even if performed by the same investigators, these factors can play a substantial role in differences and should be noted.

Response: We agree that this is an important point and added some text to the Discussion as follows:

“Additionally, even though our new approach for identifying biotinylation sites facilitated advanced identification and precise workflow (Figure 3), multiple aspects of the experiment should be carefully considered when applying or comparing it to different biological systems or techniques. Proximity labeling experiments coupled with mass spectrometry are influenced by various factors. These factors include not only experimental variables, such as the expression of proximity labeling enzyme-tagged bait proteins and cell type, but also technical considerations. Technical aspects involve the initial cell quantity, biotin labeling conditions, and optimization of protocol steps relevant to cellular labeling context. Examples of such technical considerations include determining the amount of streptavidin bead usage and selecting the appropriate bead type for pulling down biotinylated peptides.”

The graphs in Figure 4(b,c,d,e) appear to be the same graphs in S2(a,b,d,h) - why show them twice?

Response: Supplementary Figure 2 a, b, d & h were removed.

MS results need to be provided for all experiments as Excel files.

Response: The mass spectrometry proteomics data have been deposited in the ProteomeXchange Consortium via the PRIDE partner repository with the dataset identifier PXD047979, and we provided them as Excel files.

Reviewer account details:

Username: reviewer_pxd047979@ebi.ac.uk

Password: xtSHafKH

Reviewer #2 (Remarks to the Author):

Shin et al. described a method for enriching biotinylated peptides derived from proximity-dependent biotinylation by combining desthiobiotin phenol, streptavidin beads, and acid elution. Using the method, they successfully identified thousands of biotinylated peptides with high specificity (~89%). Finally, Shin et al. applied the method to study the processing body (PB) and identified novel candidates such as UBAP2L.

Overall, the manuscript is well-written, and designed, although similar works about the enrichment of biotinylated peptides using avidin-relatives were reported.

Response: We thank the reviewer for this thoughtful review of our work. We provide point-by-point responses to the reviewer's critiques below.

1. Line 95: It is not clear why biotinylated peptides can be eluted from streptavidin beads using an acidic solution. Neutraavidin should be possible due to lower stability. It would be more helpful for readers to explain why acid elution is possible.

Response: We have cited literature regarding the use of an acidic solution to elute biotinylated peptide/proteins. The mechanism underlying the action of the acidic organo-aqueous solution is similar to that of glycine elution buffers, in that the acidic condition of the buffer and the high percentage of organic solvent disrupts the integrity of the protein's structure. As a result, the structural integrity of streptavidin will be significantly reduced, which releases the biotinylated peptides that are mostly soluble in many organic buffers. We have changed the text to:

"The digested sample was incubated with streptavidin beads to capture biotinylated peptides and then stringently washed. The enriched biotinylated peptides were eluted using the acidic organo-aqueous denaturation buffer¹⁴ in which the acidic condition and the high percentage of organic solvent disrupts the integrity of the protein's structure¹⁹."

Based on our optimization experiment shown in Figure S1, both neutraavidin and streptavidin can be used as acidic organo-aqueous buffers to elute captured biotinylated peptides.

2. Figure 3b: This time why did the authors use avidin but not streptavidin?

Response: We have shown that both streptavidin and neutraavidin can be used for this method, so we intended to show two different categories of enrichment method, either avidin-relative-based capture, which included both streptavidin and neutraavidin, or anti-biotin antibody.

For the experiment illustrated in Figure 3, either streptavidin or anti-biotin antibody was used. For clarity, we have changed it to streptavidin.

3. Line 149: Please provide evidence that DBP is more efficient than BP for LC/MS analysis (before reaching Figure 4).

Response: We have included a citation explaining that DBP is more efficient than BP for LC/MS analysis, which was published by us in 2017. The following quotes from Lee et al. (2017, JACS) are provided below for your convenience:

"The thioether of conventional probe, biotin-phenol (BP), in labeled peptides is easily oxidized to the sulfoxide group during the radical generation reaction, which may reduce the signal intensity of the labeled peptides and consequently compromises the identification of low-abundant or inefficiently labeled proteins. In fact, the LC-MS intensity of the oxidized BP-labeled peptides on the tyrosine residue (+377 Da) amounted up to ~20% of the BP-labeled, original peptides (+361 Da) in human cell lines (Supporting Figure S1). However, the significant sulfur oxidation of BP might be solved by using nonsulfurated biotin or desthiobiotin"

“The MS signal enhancement after enrichment in DBP-labeling over BP-labeling was observed in APEX2-labeled human cell line sample. The MS signal gain of DBP-labeled peptides was ranging up to >10-fold higher than that of BP labeled peptides for ~80% of labeled peptides and was not dependent on protein abundance (Supporting Figure S14a). Overall, the DBP-labeling clearly resulted in a higher number of identified spectra (+35–97%), and unique labeled sites (+41–112%) than BP-labeling method in human cell lines (Supporting Figure S14b).”

4. The authors should cite, compare, and discuss the relevant papers where avidin-relatives were used to enrich biotinylated peptides;

Schiapparelli, L.M., McClatchy, D.B., Liu, H.-H., Sharma, P., Yates, J.R., 3rd, and Cline, H.T. (2014) Direct detection of biotinylated proteins by mass spectrometry. *J. Proteome Res.* 13, 3966–3978

Prikas, E., Poljak, A., and Ittner, A. (2020) Mapping p38 α mitogen-activated protein kinase signaling by proximity-dependent labeling. *Protein Sci.* 29, 1196–1210

Kohei Nishino, Harunori Yoshikawa, Kou Motani, and Hidetaka Kosako (2022) Optimized Workflow for Enrichment and Identification of Biotinylated Peptides Using Tamavidin 2-REV for BioID and Cell Surface Proteomics. *J. Proteome Res.*

Response: We have cited all the papers that the reviewer suggested, which used avidin-relatives to enrich biotinylated peptides, and we added a paragraph to the introduction and discussion sections as follows:

“As separating unlabeled contaminants from the labeled protein subpopulation during LC-MS analysis using the conventional approach is technically demanding, various techniques to identify biotinylation sites from biotin-tagged proteins have been developed that detect biotinylated peptides¹³⁻¹⁶. These techniques either focus on optimizing the protocol to enhance the enrichment of biotinylated peptides or utilize an engineered avidin-like protein called Tamavidin 2-Rev.” [Schiapparelli et al., 2014, Prikas et al., 2020, Nishino et al., 2022]

“current approaches (e.g., Spot-ID¹⁷, avidin-relative approaches^{13,15}, and anti-biotin antibody-based (Ab)^{16,18} capture) are technically demanding and result in low yields.” [Prikas et al., 2020, Nishino et al., 2022]

Method part5. 150 μ L of streptavidin beads sounds like a large volume to me. Is this for one experiment?

Response: Yes, for the clarity of bead usage, as written in the method section, we used 150 μ L of roughly 50% slurry (equivalent to total ~1.5 mg of bead, Thermo Fisher; CAT: 88817). The reason for this is that, even though cells were washed three times after the DBP labeling and subjected to acetone precipitation with sequential washes, we could still detect a certain level of unwashed free DBP that was captured/eluted along with the biotinylated peptides.

We consider that, depending on the level of expertise of experiment/bait and expression level of APEX2/TurboID-tagged bait, the amount of streptavidin usage can be further optimized.

6. Which version (year/month) of the SwissProt database was used?

Response : We added this information to the method section (2017-08-06).

7. Please add information about the mass of the (desthio)biotin label of a lysine residue.

Response: We added this information to the method section (Biotin; +226.0775; on lysine).

Reviewer #3 (Remarks to the Author):

Comments to the Author

Systematical mapping of spatial protein localization and protein-protein interactions in cells is the central task of cell biology. These data provide valuable information about almost all vital processes in cells, such as DNA replication, gene transcription, and translation, cell-cycle control and proliferation, signal transduction, and cell-cell communication. Proximity labeling techniques such as APEX and Turbo-ID, based on using engineered ascorbate peroxidase and biotin ligase mutants are now widely used for the study of intracellular phenomena. The authors Sanghee Shin et al present a work where they applied a super-resolution proximity labeling method with enhanced direct identification of biotinylation sites.

This work is a continuation of the previously published authors' method Spot-ID using desthiobiotin-phenol1 but also original with some improvements and simplification of experimental design. The authors developed a new super-resolution proximity labeling method that directly reveals the biotinylation sites based on peptide-level enrichment. In comparison with the conventional ratiometric approach utilizing protein level enrichment, the new method outperformed in terms of simpler experimental design, data reliability, LC-MS utility, and mostly unbiased identification of biotinylated proteins. The manuscript does not have technical or conceptual flaws which prohibits its publication. I would recommend this review for publication in Communications Biology with some edits, which I feel will improve the clarity of the work for the broader readership.

Response: We thank the reviewer appreciating our work and for providing valuable comments to improve the manuscript. We provide point-by-point responses to the reviewer's critiques below.

1. Did the authors check the difference in elution efficiency of DBP and BP-modified proteins and peptides from the SA-beads using an acidic organo-aqueous denaturation buffer? If it is a known elution method, please add a citation. Did the authors compare the efficiency of this elution with traditional elution using boiling in Laemmli buffer with 1-3% of SDS, 3 on protein (Western blotting) or peptide (PRM/MRM) level? (Line 95 "The enriched biotinylated peptides were eluted using acidic organo-aqueous denaturation buffer"; In section Enrichment of biotinylated peptides (Supplementary information) "To eluted biotinylated peptides, elution buffer [80% acetonitrile (Sigma-Aldrich, 900667), 0.2% TFA (Sigma-Aldrich, T6508), and 0.1% formic acid (Thermoscientific, 28905)] was added and incubated at 60°C for 5 min.")

Figure 4f

Figure 4g

Figure 4f. dot blot result of comparing different elution buffers. Figure 4g. quantitative comparison of dot blot result of eluted samples from figure 4f and supplementary figure 2g, n=4.

Response: The use of the acidic organo-aqueous buffer to elute biotin-modified peptides is known, and we have cited relevant papers. However, none of these studies compared the efficiency of this

elution technique with that of traditional elution with boiling in Laemmli buffer. We compared the elution efficiency using western blotting (dot blot), and we have described the following in the results section:

“We further crosschecked the elution efficiency of the acidic organo-aqueous buffer against that of the popular Laemmli buffer (2% SDS) based on the dot blot analysis with streptavidin-HRP due to the incompatibility of the Laemmli buffer for direct LC-MS analysis. The DBP-labeled peptide samples (by MTS-APEX2) enriched on streptavidin beads were used for the elution comparison. The results clearly showed that the acidic organo-aqueous buffer used in this study was significantly more efficient at elution than the Laemmli buffer (Fig. 4f, g, and Supplementary Fig. 2g), in addition to being LC-MS compatibility.”

2. The authors studied the topology of the mitochondrial matrix, inner membrane, and intermembrane space proteome via proximity labeling using APEX2 construct. But did they also confirm these data by using MTS-Turbo-ID and SCO1-Turbo-ID? (Line 281. "However, under several circumstances, topology determination by proximity labeling might also be restricted, as in the case of failure to biotinylate lysine or tyrosine residues that are not available for labeling, owing to extensive PTM or limited surface exposure resulting from structural considerations.")

Response: [REDACTED]

3. Please submit your LC-MS/MS raw data to proteome repository Pride, Peptide Atlas, or other⁴ and include accession number in data availability section of the manuscript.

Response: The mass spectrometry proteomics data have been deposited to the ProteomeXchange Consortium via the PRIDE partner repository with the dataset identifier PXD047979, and we provided them as Excel files.

Reviewer account details:

Username: reviewer_pxd047979@ebi.ac.uk

Password: xtSHafKH

4. Please add the meaning of the abbreviation and a brief explanation of the choice of this peptide sequence MTS. (Line 107. MTS-APEX ->24-amino-acid mitochondrial targeting sequence (MTS) from COX4 to localize APEX2 throughout the entire mitochondrial matrix⁵.)

Response: Corrected as below:

"mitochondrial matrix proteins of human embryonic kidney cells are labeled with APEX2 (mitochondrial targeting sequence with 24 amino acids, MTS-APEX2)".

5. Line 110. NC negative control

Response: We have corrected this.

6. Please add legend to lower figure topology mapping as "This paper" in Fig. 5e.

Response: We have added the legend.

REFERENCES

1. Lee, S.Y. et al. Architecture Mapping of the Inner Mitochondrial Membrane Proteome by Chemical Tools in Live Cells. *J Am Chem Soc* 139, 3651-3662 (2017).
2. de Boer, E. et al. Efficient biotinylation and single-step purification of tagged transcription factors in mammalian cells and transgenic mice. *Proc Natl Acad Sci U S A* 100, 7480-7485 (2003).
3. Villasenor, R. et al. ChromID identifies the protein interactome at chromatin marks. *Nat Biotechnol* 38, 728-736 (2020).
4. Deutsch, E.W. et al. The ProteomeXchange consortium in 2020: enabling 'big data' approaches in proteomics. *Nucleic Acids Res* 48, D1145-D1152 (2020).
5. Han, S. et al. Proximity Biotinylation as a Method for Mapping Proteins Associated with mtDNA in Living Cells. *Cell Chem Biol* 24, 404-414 (2017).

REFERENCES

- 1 Lee, S.-Y. *et al.* Architecture Mapping of the Inner Mitochondrial Membrane Proteome by Chemical Tools in Live Cells. *Journal of the American Chemical Society* **139**, 3651-3662 (2017). <https://doi.org:10.1021/jacs.6b10418>
- 2 Prikas, E., Poljak, A. & Ittner, A. Mapping p38 α mitogen-activated protein kinase signaling by proximity-dependent labeling. *Protein Sci* **29**, 1196-1210 (2020). <https://doi.org:10.1002/pro.3854>
- 3 Schiapparelli, L. M. *et al.* Direct detection of biotinylated proteins by mass spectrometry. *J Proteome Res* **13**, 3966-3978 (2014). <https://doi.org:10.1021/pr5002862>
- 4 Nishino, K., Yoshikawa, H., Motani, K. & Kosako, H. Optimized Workflow for Enrichment and Identification of Biotinylated Peptides Using Tamavidin 2-REV for BioID and Cell Surface Proteomics. *Journal of Proteome Research* **21**, 2094-2103 (2022). <https://doi.org:10.1021/acs.jproteome.2c00130>
- 5 Kim, D. I. *et al.* BioSITE: A Method for Direct Detection and Quantitation of Site-Specific Biotinylation. *J Proteome Res* **17**, 759-769 (2018). <https://doi.org:10.1021/acs.jproteome.7b00775>
- 6 Asakura, T., Adachi, K. & Schwartz, E. Stabilizing effect of various organic solvents on protein. *J Biol Chem* **253**, 6423-6425 (1978).
- 7 Udeshi, N. D. *et al.* Antibodies to biotin enable large-scale detection of biotinylation sites on proteins. *Nature methods* **14**, 1167-1170 (2017). <https://doi.org:10.1038/nmeth.4465>
- 8 Lee, S.-Y. *et al.* APEX Fingerprinting Reveals the Subcellular Localization of Proteins of Interest. *Cell Reports* **15**, 1837-1847 (2016). <https://doi.org:10.1016/j.celrep.2016.04.064>
- 9 Wakana, Y. *et al.* Bap31 is an itinerant protein that moves between the peripheral endoplasmic reticulum (ER) and a juxtannuclear compartment related to ER-associated Degradation. *Mol Biol Cell* **19**, 1825-1836 (2008). <https://doi.org:10.1091/mbc.e07-08-0781>
- 10 Annaert, W. G., Becker, B., Kistner, U., Reth, M. & Jahn, R. Export of Cellubrevin from the Endoplasmic Reticulum Is Controlled by BAP31. *Journal of Cell Biology* **139**, 1397-1410 (1997). <https://doi.org:10.1083/jcb.139.6.1397>